# Variance-Aware Regret Bounds for Stochastic Contextual Dueling Bandits

**Qiwei Di**[1*], **Tao Jin**[2*], **Yue Wu**[1], **Heyang Zhao**[1], **Farzad Farnoud**[2†], **Quanquan Gu**[1†]

[1]Department of Computer Science, University of California, Los Angeles

[2]Department of Computer Science, University of Virginia

`qiwei2000@cs.ucla.edu, taoj@virginia.edu, ywu@cs.ucla.edu,`
`hyzhao@cs.ucla.edu,farzad@virginia.edu,qgu@cs.ucla.edu`

## Abstract

Dueling bandits is a prominent framework for decision-making involving preferential feedback, a valuable feature that fits various applications involving human interaction, such as ranking, information retrieval, and recommendation systems. While substantial efforts have been made to minimize the cumulative regret in dueling bandits, a notable gap in the current research is the absence of regret bounds that account for the inherent uncertainty in pairwise comparisons between the dueling arms. Intuitively, greater uncertainty suggests a higher level of difficulty in the problem. To bridge this gap, this paper studies the problem of contextual dueling bandits, where the binary comparison of dueling arms is generated from a generalized linear model (GLM). We propose a new SupLinUCB-type algorithm that enjoys computational efficiency and a variance-aware regret bound $\widetilde{O}\big(d\sqrt{\sum_{t=1}^{T}\sigma_t^2}+d\big)$, where $\sigma_t$ is the variance of the pairwise comparison in round $t$, $d$ is the dimension of the context vectors, and $T$ is the time horizon. Our regret bound naturally aligns with the intuitive expectation — in scenarios where the comparison is deterministic, the algorithm only suffers from an $\widetilde{O}(d)$ regret. We perform empirical experiments on synthetic data to confirm the advantage of our method over previous variance-agnostic algorithms.

## 1 Introduction

The multi-armed bandit (MAB) model has undergone comprehensive examination as a framework for decision-making with uncertainty. Within this framework, an agent has to select one specific *"arm"* to pull in each round, and receives a *stochastic* reward as feedback. The objective is to maximize the cumulative reward accumulated over all rounds. While the MAB model provides a robust foundation for various applications, the reality is that many real-world tasks present an intractably large action space coupled with intricate contextual information. Consequently, this challenge has led to the proposal of the (linear) contextual bandit model, where the reward is intricately linked to both the context associated with the selected arm and the underlying reward function. A series of work into the linear contextual bandits has led to efficient algorithms such as LinUCB (Li et al., 2010; Chu et al., 2011) and OFUL (Abbasi-Yadkori et al., 2011).

In scenarios where feedback is based on subjective human experiences – a phenomenon evident in fields such as information retrieval (Yue & Joachims, 2009), ranking (Minka et al., 2018), crowd-sourcing (Chen et al., 2013), and Reinforcement Learning from Human Feedback (RLHF) (Ouyang et al., 2022) – preferential choices emerge as a more natural and intuitive form of feedback compared with numerical evaluations. The rationale behind preference feedback lies in the fact that numerical scores can exhibit significant variability among individuals, resulting in noisy and poorly calibrated rewards. On the contrary, a binary signal from preferential feedback remains independent of scale and is thus more reliable. This distinction gives rise to a specialized variant of the MAB problem known as *dueling bandits* (Yue et al., 2012). In this setting, the agent simultaneously pulls two arms and receives binary preferential feedback, which essentially indicates the outcome of a comparison between the chosen arms. A line of works proposed efficient and practical algorithms for multi-armed dueling bandits based on upper confidence bound (UCB) (Zoghi et al., 2014; 2015) or Thompson

---

*Equal Contribution

†Co-corresponding Authors

sampling (Wu & Liu, 2016). Similar to linear contextual bandits, considerable effort has been invested in developing efficient algorithms that minimize the cumulative regret for the contextual dueling bandits (Saha, 2021; Bengs et al., 2022).

Intuitively, the variance of the noise in the feedback signal determines the difficulty of the problem. To illustrate, consider an extreme case, where the feedback of a linear contextual bandit is noiseless (i.e., the variance is zero). A learner can recover the underlying reward function precisely by exploring each dimension only once, and suffer a $\widetilde{O}(d)$ regret in total, where $d$ is the dimension of the context vector. This motivates a series of works on establishing variance-aware regret bounds for multi-armed bandits, e.g. (Audibert et al., 2009; Mukherjee et al., 2017) and contextual bandits, e.g. (Zhou et al., 2021; Zhang et al., 2021b; Kim et al., 2022; Zhao et al., 2023b;a). This observation also remains valid when applied to the dueling bandit scenario. In particular, the binary preferential feedback is typically assumed to adhere to a Bernoulli distribution, with the mean value denoted by $p$. The variance reaches its maximum when $p$ is close to $1/2$, a situation that is undesirable in human feedback applications, as it indicates a high level of disagreement or indecision. Therefore, maintaining a low variance in comparisons is usually preferred, and variance-dependent dueling algorithms are desirable because they can potentially perform better than those algorithms that only have worst-case regret guarantees. This leads to the following research question:

*Can we design a dueling bandit algorithm with a variance-aware regret bound?*

We give an affirmative answer to this question by studying the dueling bandit problem with a contextualized generalized linear model, which is in the same setting as Saha (2021); Bengs et al. (2022). We summarize our contributions as follows:

- We propose a new algorithm, named `VACDB`, to obtain a variance-aware regret guarantee. This algorithm is built upon several innovative designs, including (1) adaptation of multi-layered estimators to generalized linear models where the mean and variance are coupled (i.e., Bernoulli distribution), (2) symmetric arm selection that naturally aligns with the actual reward maximization objective in dueling bandits.

- We prove that our algorithm enjoys a variance-aware regret bound $\widetilde{O}\big(d\sqrt{\sum_{t=1}^{T}\sigma_t^2} + d\big)$, where $\sigma_t$ is the variance of the comparison in round $t$. Our algorithm is computationally efficient and does not require any prior knowledge of the variance level, which is available in the dueling bandit scenario. In the deterministic case, our regret bound becomes $\widetilde{O}(d)$, showcasing a remarkable improvement over previous works. When the variances of the pairwise comparison are the same across different pairs of arms, our regret reduces to the worst-case regret of $\widetilde{O}(d\sqrt{T})$, which matches the lower bound $\Omega(d\sqrt{T})$ proved in Bengs et al. (2022)

- We compare our algorithm with many strong baselines on synthetic data. Our experiments demonstrate the empirical advantage of the proposed algorithm in terms of regret and adaptiveness when faced with environments with varying variances.

- As an additional outcome of our research, we identified an unrigorous argument in the existing analysis of MLE for generalized linear bandits. To rectify this issue, we provide a rigorous proof based on Brouwer's invariance of domain property (Brouwer, 1911), which is discussed further in Appendix D.

**Notation** In this paper, we use plain letters such as $x$ to denote scalars, lowercase bold letters such as $\mathbf{x}$ to denote vectors and uppercase bold letters such as $\mathbf{X}$ to denote matrices. For a vector $\mathbf{x}$, $\|\mathbf{x}\|_2$ denotes its $\ell_2$-norm. The weighted $\ell_2$-norm associated with a positive-definite matrix $\mathbf{A}$ is defined as $\|\mathbf{x}\|_{\mathbf{A}} = \sqrt{\mathbf{x}^\top \mathbf{A}\mathbf{x}}$. For two symmetric matrices $\mathbf{A}$ and $\mathbf{B}$, we use $\mathbf{A} \succeq \mathbf{B}$ to denote $\mathbf{A} - \mathbf{B}$ is positive semidefinite. We use $\mathbb{1}$ to denote the indicator function and $\mathbf{0}$ to denote the zero vector. For a positive integer $N$, we use $[N]$ to denote $\{1, 2, \ldots, N\}$. We use $\mathbf{x}_{1:t}$ to denote the set $\{\mathbf{x}_i\}_{1 \le i \le t}$. We use standard asymptotic notations including $O(\cdot), \Omega(\cdot), \Theta(\cdot)$, and $\widetilde{O}(\cdot), \widetilde{\Omega}(\cdot), \widetilde{\Theta}(\cdot)$ will hide logarithmic factors.

## 2 RELATED WORK

**Multi-Armed Bandits and Contextual Bandits.** The multi-armed bandit problem involves an agent making sequential decisions among multiple arms based on the observation of stochastic reward, with the goal of maximizing the cumulative rewards over time. It has been widely studied, including works such as Lai et al. (1985); Lai (1987); Auer (2002); Auer et al. (2002); Kalyanakrishnan et al.

(2012); Lattimore & Szepesvári (2020); Agrawal & Goyal (2012). To deal with large decision spaces with potentially infinitely many actions or to utilize contextual information, extensive studies have been conducted in contextual bandits. Some work focused on contextual linear bandits, where the mean reward of an arm is a linear function of some feature vectors, including algorithms such as LinUCB/SupLinUCB (Chu et al., 2011), OFUL (Abbasi-Yadkori et al., 2011). Other works, such as (Filippi et al., 2010; Li et al., 2017; Jun et al., 2017), studied the generalized linear bandits where the mean reward is from a generalized linear model (GLM).

**Dueling Bandits.** The problem of dueling bandits is a variant of the multi-armed bandits, where the stochastic reward is replaced by a pairwise preference. This model was first proposed in Yue et al. (2012). Many works (Zoghi et al., 2014; Komiyama et al., 2015) studied this problem, assuming the existence of a Condorcet winner, which is one arm that beats all the other arms. There are also works on other types of winners such as Copeland winner (Zoghi et al., 2015; Wu & Liu, 2016; Komiyama et al., 2016), Borda winner (Jamieson et al., 2015; Falahatgar et al., 2017; Heckel et al., 2018; Saha et al., 2021; Wu et al., 2023) and von Neumann winner (Ramamohan et al., 2016; Dudík et al., 2015; Balsubramani et al., 2016). Similar to the idea of contextual bandits, some works considered regret minimization for dueling bandits with context information. Kumagai (2017) studied the contextual dueling bandit problem where the feedback is based on a cost function. They proposed a stochastic mirror descent algorithm and proved the regret upper bound under strong convexity and smoothness assumptions. Saha (2021) proposed algorithms and lower bounds for contextual preference bandits with logistic link function, considering pairwise and subsetwise preferences, respectively. Bengs et al. (2022) further extended to the contextual linear stochastic transitivity model, allowing arbitrary comparison function, and provided efficient algorithms along with a matching lower bound for the weak regret. For a recent comprehensive survey of dueling bandits, please refer to Bengs et al. (2021). Our work studies the same model as Saha (2021); Bengs et al. (2021).

**Variance-Aware Bandits.** It has been shown empirically that leveraging variance information in multi-armed bandit algorithms can enjoy performance benefits (Auer et al., 2002). In light of this, Audibert et al. (2009) proposed an algorithm, named `UCBV`, which is based on Bernstein's inequality equipped with empirical variance. It provided the first analysis of variance-aware algorithms, demonstrating an improved regret bound. `EUCBV` Mukherjee et al. (2017) is another variance-aware algorithm that employs an elimination strategy. It incorporates variance estimates to determine the confidence bounds of the arms. For linear bandits, Zhou et al. (2021) proposed a Bernstein-type concentration inequality for self-normalized martingales and designed an algorithm named `Weighted OFUL`. This approach used a weighted ridge regression scheme, using variance to discount each sample's contribution to the estimator. In particular, they proved a variance-dependent regret upper bound, which was later improved by Zhou & Gu (2022). These two works assumed the knowledge of variance information. Without knowing the variances, Zhang et al. (2021a) and Kim et al. (2022) obtained the variance-dependent regret bound by constructing variance-aware confidence sets. (Zhao et al., 2023b) proposed an algorithm named `MOR-UCB` with the idea of partitioning the observed data into several layers and grouping samples with similar variance into the same layer. A similar idea was used in Zhao et al. (2023a) to design a SupLin-type algorithm `SAVE`. It assigns collected samples to $L$ layers according to their estimated variances, where each layer has twice the variance upper bound as the one at one level lower. In this way, for each layer, the estimated variance of one sample is at most twice as the others. Their algorithm is computationally tractable with a variance-dependent regret bound based on a Freedman-type concentration inequality and adaptive variance-aware exploration.

## 3 PROBLEM SETUP

In this work, we consider a preferential feedback model with contextual information. In this model, an agent learns through sequential interactions with its environment over a series of rounds indexed by $t$, where $t \in [T]$ and $T$ is the total number of rounds. In each round $t$, the agent is presented with a finite set of alternatives, with each alternative being characterized by its associated feature in the contextual set $\mathcal{A}_t \subseteq \mathbb{R}^d$. Following the convention in bandit theory, we refer to these alternatives as *arms*. Both the number of alternatives and the contextual set $\mathcal{A}_t$ can vary with the round index $t$. Afterward, the agent selects a pair of arms, with features $(\mathbf{x}_t, \mathbf{y}_t)$ respectively. The environment then compares the two selected arms and returns a stochastic feedback $o_t$, which takes a value from the set $\{0, 1\}$. This feedback informs the agent which arm is preferred: When $o_t = 1$ (resp. $o_t = 0$), the arm with feature $\mathbf{x}_t$ (resp. $\mathbf{y}_t$) wins.

We assume that stochastic feedback $o_t$ follows a Bernoulli distribution, where the expected value $p_t$ is determined by a generalized linear model (GLM). To be more specific, let $\mu(\cdot)$ be a fixed link

function that is increasing monotonically and satisfies $\mu(x) + \mu(-x) = 1$. We assume the existence of an *unknown* parameter $\boldsymbol{\theta}^* \in \mathbb{R}^d$ which generates the preference probability when two contextual vectors are given, i.e.

$$\mathbb{P}(o_t = 1) = \mathbb{P}(\text{arm with } \mathbf{x}_t \text{ is preferred over arm with } \mathbf{y}_t) = p_t = \mu((\mathbf{x}_t - \mathbf{y}_t)^\top \boldsymbol{\theta}^*).$$

This model is the same as the linear stochastic transitivity (LST) model in Bengs et al. (2022), which includes the Bradley-Terry-Luce (BTL) model (Hunter, 2003; Luce, 1959), Thurstone-Mosteller model (Thurstone, 1994) and the exponential noise model as special examples. Please refer to Bengs et al. (2022) for details. The preference model studied in Saha (2021) can be treated as a special case where the link function is logistic.

We make the assumption on the boundness of the true parameter $\boldsymbol{\theta}^*$ and the feature vector.

**Assumption 3.1.** $\|\boldsymbol{\theta}^*\|_2 \leq 1$. There exists a constant $A > 0$ such that for all $t \in [T]$ and all $\mathbf{x} \in \mathcal{A}_t$, $\|\mathbf{x}\|_2 \leq A$.

Additionally, we make the following assumption on the link function $\mu$, which is common in the study of generalized linear contextual bandits (Filippi et al., 2010; Li et al., 2017).

**Assumption 3.2.** The link function $\mu$ is differentiable. Furthermore, the first derivative $\dot{\mu}$ satisfies $\kappa_\mu \leq \dot{\mu}(\cdot) \leq L_\mu$ for some constants $L_\mu, \kappa_\mu > 0$.

We define the random noise $\epsilon_t = o_t - p_t$. Since the stochastic feedback $o_t$ adheres to the Bernoulli distribution with expected value $p_t$, $\epsilon_t \in \{-p_t, 1 - p_t\}$. From the definition of $\epsilon_t$, we can see that $|\epsilon_t| \leq 1$. Furthermore, we make the following assumptions:

$$\mathbb{E}[\epsilon_t | \mathbf{x}_{1:t}, \mathbf{y}_{1:t}, \epsilon_{1:t-1}] = 0, \mathbb{E}[\epsilon_t^2 | \mathbf{x}_{1:t}, \mathbf{y}_{1:t}, \epsilon_{1:t-1}] = \sigma_t^2.$$

Intuitively, $\sigma_t$ reflects the difficulty associated with comparing the two arms:

- When $p_t$ is around $1/2$, it suggests that the arms are quite similar, making the comparison challenging. Under this circumstance, the variance $\sigma_t$ tends toward a constant, reaching a maximum value of $1/4$.

- On the contrary, as $p_t$ approaches 0 or 1, it signals that one arm is distinctly preferable over the other, thus simplifying the comparison. In such scenarios, the variance $\sigma_t$ decreases significantly toward 0.

The learning objective is to minimize the cumulative average regret defined as

$$\text{Regret}(T) = \frac{1}{2} \sum_{t=1}^T \left[ 2\mathbf{x}_t^{*\top} \boldsymbol{\theta}^* - (\mathbf{x}_t + \mathbf{y}_t)^\top \boldsymbol{\theta}^* \right], \tag{3.1}$$

where $\mathbf{x}_t^* = \arg\max_{\mathbf{x} \in \mathcal{A}_t} \mathbf{x}^\top \boldsymbol{\theta}^*$ is the contextual/feature vector of the optimal arm in round $t$. This definition is the same as the average regret studied in (Saha, 2021; Bengs et al., 2022). Note that in Bengs et al. (2022), besides the average regret, they also studied another type of regret, called weak regret. Since the weak regret is smaller than the average regret, the regret bound proved in our paper can immediately imply a regret bound defined by the weak regret.

# 4 ALGORITHM

## 4.1 OVERVIEW OF THE ALGORITHM

In this section, we present our algorithm named `VACDB` in Algorithm 1. Our algorithm shares a similar structure with `Sta'D` in Saha (2021) and `SupCoLSTIM` in Bengs et al. (2022). The core of our algorithm involves a sequential arm elimination process: from Line 6 to Line 18, our algorithm conducts arm selection with a layered elimination procedure. Arms are progressively eliminated across layers, with increased exploration precision in the subsequent layers. Starting at layer $\ell = 1$, our algorithm incorporates a loop comprising three primary conditional phases: Exploitation (Lines 7-9), Elimination (Lines 10-12) and Exploration (Lines 14-16). When all arm pairs within a particular layer have low uncertainty, the elimination procedure begins, dropping the arms with suboptimal estimated values. This elimination process applies an adaptive bonus radius based on variance information. A more comprehensive discussion can be found in Section 4.3. Subsequently, it advances to a higher layer, where exploration is conducted over the eliminated set. Upon encountering a layer with arm pairs of higher uncertainty than desired, our algorithm explores them and receives the feedback. Once comprehensive exploration has been achieved across layers and the uncertainty for all remaining arm pairs is small enough, our algorithm leverages the estimated

---

**Algorithm 1** Variance-Aware Contextual Dueling Bandit (`VACDB`)

---

1: **Require:** $\alpha > 0$, $L \leftarrow \lceil \log_2(1/\alpha) \rceil$, $\kappa_\mu$, $L_\mu$.
2: **Initialize:** For $\ell \in [L]$, $\widehat{\boldsymbol{\Sigma}}_{1,\ell} \leftarrow 2^{-2\ell}\mathbf{I}$, $\widehat{\boldsymbol{\theta}}_{1,\ell} \leftarrow \mathbf{0}$, $\boldsymbol{\Psi}_{1,\ell} \leftarrow \emptyset$, $\widehat{\beta}_{1,\ell} \leftarrow 2^{-\ell}(1 + 1/\kappa_\mu)$
3: **for** $t = 1, \ldots, T$ **do**
4:     Observe $\mathcal{A}_t$
5:     Let $\mathcal{A}_{t,1} \leftarrow \mathcal{A}_t$, $\ell \leftarrow 1$.
6:     **while** $\mathbf{x}_t, \mathbf{y}_t$ are not specified **do**
7:         **if** $\|\mathbf{x}_t - \mathbf{y}_t\|_{\widehat{\boldsymbol{\Sigma}}_{t,\ell}^{-1}} \leq \alpha$ for all $\mathbf{x}_t, \mathbf{y}_t \in \mathcal{A}_{t,\ell}$ **then**
8:             Choose $\mathbf{x}_t, \mathbf{y}_t = \text{argmax}_{\mathbf{x},\mathbf{y} \in \mathcal{A}_{t,\ell}} \left\{ (\mathbf{x} + \mathbf{y})^\top \widehat{\boldsymbol{\theta}}_{t,\ell} + \widehat{\beta}_{t,\ell} \|\mathbf{x} - \mathbf{y}\|_{\widehat{\boldsymbol{\Sigma}}_{t,\ell}^{-1}} \right\}$
            and observe $o_t = \mathbb{1}(\mathbf{x}_t \succ \mathbf{y}_t)$       `//Exploitation (Lines 7-9)`
9:             Keep the same index sets at all layers: $\boldsymbol{\Psi}_{t+1,\ell'} \leftarrow \boldsymbol{\Psi}_{t,\ell'}$ for all $\ell' \in [L]$
10:        **else if** $\|\mathbf{x}_t - \mathbf{y}_t\|_{\widehat{\boldsymbol{\Sigma}}_{t,\ell}^{-1}} \leq 2^{-\ell}$ for all $\mathbf{x}_t, \mathbf{y}_t \in \mathcal{A}_{t,\ell}$ **then**
11:             $\mathcal{A}_{t,\ell+1} \leftarrow \left\{ \mathbf{x} \in \mathcal{A}_{t,\ell} \mid \mathbf{x}^\top \widehat{\boldsymbol{\theta}}_{t,\ell} \geq \max_{\mathbf{x}' \in \mathcal{A}_{t,\ell}} {\mathbf{x}'}^\top \widehat{\boldsymbol{\theta}}_{t,\ell} - 2^{-\ell}\widehat{\beta}_{t,\ell} \right\}$
12:             $\ell = \ell + 1$                       `//Elimination (Lines 10-12)`
13:        **else**
14:             Choose $\mathbf{x}_t, \mathbf{y}_t$ such that $\|\mathbf{x}_t - \mathbf{y}_t\|_{\widehat{\boldsymbol{\Sigma}}_{t,\ell}^{-1}} > 2^{-\ell}$
            and observe $o_t = \mathbb{1}(\mathbf{x}_t \succ \mathbf{y}_t)$       `//Exploration (Lines 14-16)`
15:             Compute the weight $w_t \leftarrow 2^{-\ell}/\|\mathbf{x}_t - \mathbf{y}_t\|_{\widehat{\boldsymbol{\Sigma}}_{t,\ell}^{-1}}$
16:             Update the index sets $\boldsymbol{\Psi}_{t+1,\ell} \leftarrow \boldsymbol{\Psi}_{t,\ell} \cup \{t\}$ and $\boldsymbol{\Psi}_{t+1,\ell'} \leftarrow \boldsymbol{\Psi}_{t,\ell'}$ for all $\ell' \in [L]/\{\ell\}$
17:        **end if**
18:     **end while**
19:     For $\ell \in [L]$ such that $\boldsymbol{\Psi}_{t+1,\ell} \neq \boldsymbol{\Psi}_{t,\ell}$, update $\widehat{\boldsymbol{\Sigma}}_{t+1,\ell} \leftarrow \widehat{\boldsymbol{\Sigma}}_{t,\ell} + w_t^2(\mathbf{x}_t - \mathbf{y}_t)(\mathbf{x}_t - \mathbf{y}_t)^\top$
20:     Calculate the MLE $\widehat{\boldsymbol{\theta}}_{t+1,\ell}$ by solving the equation:

$$2^{-2\ell}\kappa_\mu\boldsymbol{\theta} + \sum_{s \in \boldsymbol{\Psi}_{t+1,\ell}} w_s^2 \Big( \mu\big((\mathbf{x}_s - \mathbf{y}_s)^\top \boldsymbol{\theta}\big) - o_s \Big)(\mathbf{x}_s - \mathbf{y}_s) = \mathbf{0}$$

21:     Compute $\widehat{\beta}_{t+1,\ell}$ according to (4.3)
22:     For $\ell \in [L]$ such that $\boldsymbol{\Psi}_{t+1,\ell} = \boldsymbol{\Psi}_{t,\ell}$, let $\widehat{\boldsymbol{\Sigma}}_{t+1,\ell} = \widehat{\boldsymbol{\Sigma}}_{t,\ell}$, $\widehat{\boldsymbol{\theta}}_{t+1,\ell} \leftarrow \widehat{\boldsymbol{\theta}}_{t,\ell}$, $\widehat{\beta}_{t+1,\ell} \leftarrow \widehat{\beta}_{t,\ell}$
23: **end for**

---

parameters in the last layer to select the best arm from the remaining arms. For a detailed discussion of the selection policy, please refer to Section 4.4. After arm selection in the exploration phase, the estimator of the current layer is updated (Lines 19-22) using the regularized MLE, which will be discussed in more details in Section 4.2. Note that our algorithm maintains an index set $\Psi_{t,\ell}$ for each layer, comprising all rounds before round $t$ when the algorithm conducts exploration in layer $\ell$. As a result, for each exploration step, only one of the estimators $\widehat{\boldsymbol{\theta}}_{t,\ell}$ needs to be updated. Furthermore, our algorithm updates the covariance matrix $\widehat{\boldsymbol{\Sigma}}_{t,\ell}$ used to estimate uncertainty (Line 19).

## 4.2 REGULARIZED MLE

Most of the previous work adopted standard MLE techniques to maintain an estimator of $\boldsymbol{\theta}^*$ in the generalized linear bandit model (Filippi et al., 2010; Li et al., 2017), which requires an initial exploration phase to ensure a balanced input dataset across $\mathbb{R}^d$ for the MLE. In the dueling bandits setting, where the feedback in each round can be seen as a generalized linear reward, Saha (2021); Bengs et al. (2022) also applied a similar MLE in their algorithms. As a result, a random initial exploration phase is also inherited to ensure that the MLE equation has a unique solution. However, in our setting, where the decision set varies among rounds and is even arbitrarily decided by the environment, this initial exploration phase cannot be directly applied to control the minimum eigenvalue of the covariance matrix.

To resolve this issue, we introduce a regularized MLE for contextual dueling bandits, which is more well-behaved in the face of extreme input data and does not require an additional exploration phase at the starting rounds. Specifically, the regularized MLE is the solution of the following equation:

$$\lambda\boldsymbol{\theta} + \sum_s w_s^2 \Big( \mu\big((\mathbf{x}_s - \mathbf{y}_s)^\top \boldsymbol{\theta}\big) - o_s \Big)(\mathbf{x}_s - \mathbf{y}_s) = \mathbf{0}, \tag{4.1}$$

where we add the additional regularization term $\lambda \boldsymbol{\theta}$ to make sure that the estimator will change mildly. From the theoretical viewpoint, our proposed regularization term leads to a non-singularity guarantee for the covariance matrix. Additionally, we add some weights here to obtain a tighter concentration inequality. Concretely, with a suitable choice of the parameters in each layer and a Freedman-type inequality first introduced in Zhao et al. (2023a), we can prove a concentration inequality for the estimator in the $\ell$-th layer:

$$\left\| \boldsymbol{\theta}^* - \widehat{\boldsymbol{\theta}}_{t,\ell} \right\|_{\widehat{\boldsymbol{\Sigma}}_{t,\ell}} \leq \frac{2^{-\ell}}{\kappa_\mu} \left[ 16 \sqrt{\sum_{s \in \boldsymbol{\Psi}_{t,\ell}} w_s^2 \sigma_s^2 \log(4t^2 L/\delta)} + 6 \log(4t^2 L/\delta) \right] + 2^{-\ell}. \quad (4.2)$$

This upper bound scales with $2^{-\ell}$, which arises from our choice of the weights.

The regularized MLE can be formulated as a finite-sum offline optimization problem. For many widely used models, such as the Bradley-Terry-Luce (BTL) model (Hunter, 2003; Luce, 1959), the regularized MLE is a strongly convex and smooth optimization problem. We can solve it using accelerated gradient descent (Nesterov, 2003) and SVRG (Johnson & Zhang, 2013), both of which achieve a linear rate of convergence. This can mitigate the scalability issues caused by the increasing number of iterations. The regularized MLE can also be solved by an online learning algorithm such as in Jun et al. (2017) and Zhao et al. (2023b), where additional effort is required for the analysis.

### 4.3 MULTI-LAYER STRUCTURE WITH VARIANCE-AWARE CONFIDENCE RADIUS

Due to the multi-layered structure of our algorithm, the construction of the confidence set is of paramount importance. Our algorithm distinguishes itself from prior multi-layered algorithms (Saha, 2021; Bengs et al., 2022) primarily through a variance-aware adaptive selection of the confidence radius, which helps to achieve a variance-aware regret bound. Intuitively, we should choose the confidence radius $\widehat{\beta}_{t,\ell}$ based on the concentration inequality (4.2). However, it depends on the true variance $\sigma_s$, of which we do not have prior knowledge. To address this issue, we estimate it using the estimator $\widehat{\boldsymbol{\theta}}_{t,\ell}$. We choose

$$\widehat{\beta}_{t,\ell} := \frac{16 \cdot 2^{-\ell}}{\kappa_\mu} \sqrt{\left( 8\widehat{\mathrm{Var}}_{t,\ell} + 18 \log(4(t+1)^2 L/\delta) \right) \log(4t^2 L/\delta)}$$

$$+ \frac{6 \cdot 2^{-\ell}}{\kappa_\mu} \log(4t^2 L/\delta) + 2^{-\ell+1}, \quad (4.3)$$

where

$$\widehat{\mathrm{Var}}_{t,\ell} := \begin{cases} \sum_{s \in \Psi_{t,\ell}} w_s^2 \left( o_s - \mu((\mathbf{x}_s - \mathbf{y}_s)^\top \widehat{\boldsymbol{\theta}}_{t,\ell}) \right)^2, & 2^\ell \geq 64(L_\mu/\kappa_\mu) \sqrt{\log(4(t+1)^2 L/\delta)}, \\ |\Psi_{t,\ell}|, & \text{otherwise.} \end{cases}$$

The varied selections of $\widehat{\mathrm{Var}}_{t,\ell}$ arise from the fact that our variance estimator becomes more accurate at higher layers. For those low layers, we employ the natural upper bound $\sigma_i \leq 1$. Note that this situation arises only $\Theta(\log \log(T/\delta))$ times, which is a small portion of the total layers $L = \Theta(\log T)$. In our proof, we deal with two cases separately. Due to the limited space available here, the full proof can be found in Appendix E.

### 4.4 SYMMETRIC ARM SELECTION

In this subsection, we focus on the arm selection policy described in Line 9. To our knowledge, this policy is new and has never been studied in prior work for the (generalized) linear dueling bandit problem. In detail, suppose that we have an estimator $\widehat{\boldsymbol{\theta}}_t$ in round $t$ that lies in a high probability confidence set:

$$\left\{ \boldsymbol{\theta} : \left\| \boldsymbol{\theta} - \boldsymbol{\theta}^* \right\|_{\widehat{\boldsymbol{\Sigma}}_t} \leq \beta_t \right\},$$

where $\widehat{\boldsymbol{\Sigma}}_t = \lambda \mathbf{I} + \sum_{i=1}^{t-1} (\mathbf{x}_i - \mathbf{y}_i)(\mathbf{x}_i - \mathbf{y}_i)^\top$. Our choice of arms can be written as

$$\mathbf{x}_t, \mathbf{y}_t = \operatorname*{argmax}_{\mathbf{x}, \mathbf{y} \in \mathcal{A}_t} \left[ (\mathbf{x} + \mathbf{y})^\top \widehat{\boldsymbol{\theta}}_t + \beta_t \|\mathbf{x} - \mathbf{y}\|_{\widehat{\boldsymbol{\Sigma}}_t^{-1}} \right]. \quad (4.4)$$

Intuitively, we utilize $(\mathbf{x} + \mathbf{y})^\top \widehat{\boldsymbol{\theta}}_t$ as the estimated score and incorporate an exploration bonus dependent on $\|\mathbf{x} - \mathbf{y}\|_{\widehat{\boldsymbol{\Sigma}}_t^{-1}}$. Our symmetric selection of arms aligns with the nature of dueling bandits where the order of arms does not matter. Here we compare it with several alternative arm selection criteria that have appeared in previous works.

The `MaxInP` algorithm in Saha (2021) builds the so-called "promising" set that includes the optimal arm:

$$\mathcal{C}_t = \left\{ \mathbf{x} \in \mathcal{A}_t \mid (\mathbf{x} - \mathbf{y})^\top \widehat{\boldsymbol{\theta}}_t + \beta_t \|\mathbf{x} - \mathbf{y}\|_{\widehat{\boldsymbol{\Sigma}}_t^{-1}} \geq 0, \forall \mathbf{y} \in \mathcal{A}_t \right\}.$$

It chooses the symmetric arm pair from the set $\mathcal{C}_t$ that has the highest pairwise score variance (maximum informative pair), i.e.,

$$\mathbf{x}_t, \mathbf{y}_t = \underset{\mathbf{x}, \mathbf{y} \in \mathcal{C}_t}{\operatorname{argmax}} \|\mathbf{x} - \mathbf{y}\|_{\boldsymbol{\Sigma}_t^{-1}}.$$

The `Sta'D` algorithm in Saha (2021) uses an asymmetric arm selection criterion, which selects the first arm with the highest estimated score, i.e.,

$$\mathbf{x}_t = \underset{\mathbf{x} \in \mathcal{A}_t}{\operatorname{argmax}} \mathbf{x}^\top \widehat{\boldsymbol{\theta}}_t.$$

Following this, it selects the second arm as the toughest competitor to the arm $\mathbf{x}_t$, with a bonus term related to $\|\mathbf{x}_t - \mathbf{y}\|_{\Sigma_t^{-1}}$, i.e.,

$$\mathbf{y}_t = \underset{\mathbf{y} \in \mathcal{A}_t}{\operatorname{argmax}} \mathbf{y}^\top \widehat{\boldsymbol{\theta}}_t + 2\beta_t \|\mathbf{x}_t - \mathbf{y}\|_{\boldsymbol{\Sigma}_t^{-1}}. \tag{4.5}$$

Similar arm selection criterion has also been used in the `CoLSTIM` algorithm (Bengs et al., 2022). We can show that these two alternative arm selection policies result in comparable regret decomposition and can establish similar regret upper bound. A more detailed analysis can be found in Appendix C.

## 5 MAIN RESULTS

### 5.1 VARIANCE-AWARE REGRET BOUND

In this section, we summarize our main results in the following theorem.

**Theorem 5.1.** If we set $\alpha = 1/(T^{3/2})$, then with probability at least $1 - 2\delta$, the regret of Algorithm 1 is bounded as

$$\text{Regret}(T) = \widetilde{O}\left( \frac{d}{\kappa_\mu} \sqrt{\sum_{t=1}^T \sigma_t^2} + d\left( \frac{L_\mu^2}{\kappa_\mu^2} + \frac{1}{\kappa_\mu} \right) \right).$$

This regret can be divided into two parts, corresponding to the regret incurred from the exploration steps (Line 14) and the exploitation steps (Line 8). The exploitation-induced regret is always $\widetilde{O}(1)$ as shown in (5.1), and thus omitted by the big-O notation. The total regret is dominated by the exploration-induced regret, which mainly depends on the total variance $\sum_{t=1}^T \sigma_t^2$. Note that the comparisons during the exploration steps only happen between non-identical arms ($\mathbf{x}_t \neq \mathbf{y}_t$).

**Remark 5.2.** To show the advantage of variance awareness, consider the extreme case where the comparisons are deterministic. More specifically, for any two arms with contextual vectors $\mathbf{x}$ and $\mathbf{y}$, the comparison between arm $\mathbf{x}$ and item $\mathbf{y}$ is determined by $o_t = \mathbb{1}\left\{ \mathbf{x}_t^\top \boldsymbol{\theta}^* > \mathbf{y}_t^\top \boldsymbol{\theta}^* \right\}$, and thus has zero variance. Our algorithm can account for the zero variance, and the regret becomes $\widetilde{O}(d)$, which is optimal since recovering the parameter $\boldsymbol{\theta}^* \in \mathbb{R}^d$ requires exploring each dimension.

**Remark 5.3.** The setting we study is quite general, where the arm set is time-varying, and therefore, the variance of arms can vary with respect to time and arms. When we restrict our setting to a special case with uniform variances for all pairwise comparisons, i.e., $\sigma_t^2 = \sigma^2$ for all $t$, our upper bound becomes $\widetilde{O}(\sigma d\sqrt{T})$. This results in a regret bound that does not depend on the random variable $\sigma_t^2$.

**Remark 5.4.** In the worst-case scenario, the variance of the arm comparison is upper bounded by $1/4$, our regret upper bound becomes $\widetilde{O}(d\sqrt{T})$, which matches the regret lower bound $\Omega(d\sqrt{T})$ for dueling bandits with exponentially many arms proved in Bengs et al. (2022), up to logarithmic factors. This regret bound also recovers the regret bounds of `MaxInP` (Saha, 2021) and `CoLSTIM` (Bengs et al., 2022). Compared with `Sta'D` (Saha, 2021) and `SupCoLSTIM` (Bengs et al., 2022), our regret bound is on par with their regret bounds provided the number of arms $K$ is large. More specifically, their regret upper bounds are $\widetilde{O}(\sqrt{dT \log K})$. When $K$ is exponential in $d$, their regret bound becomes $\widetilde{O}(d\sqrt{T})$, which is of the same order as our regret bound.

**Remark 5.5.** Notably, in Bengs et al. (2022), they made an assumption that the context vectors can span the total $d$-dimensional Euclidean space, which is essential in their initial exploration phase. In our work, we replace the initial exploration phase with a regularizer, thus relaxing their assumption.

## 5.2 Proof Sketch of Theorem 5.1

As we describe in Section 4, the arm selection is specified in two places, the exploration part (Lines 14 - 16) and the exploitation part (Lines 8 - 9). Given the update rule of the index set, each step within the exploration part will be included by the final index set $\Psi_{T+1,\ell}$ of a singular layer $\ell$. Conversely, steps within the exploitation part get into $T/\cup_{\ell\in[L]}\Psi_{T+1,\ell}$. Using this division, we can decompose the regret into :

$$\text{Regret}(T) = \frac{1}{2}\Bigg[\underbrace{\sum_{s\in[T]/(\cup_{\ell\in[L]}\Psi_{T+1,\ell})}\Big(2\mathbf{x}_s^{*\top}\boldsymbol{\theta}^* - (\mathbf{x}_s^{\top}\boldsymbol{\theta}^* + \mathbf{y}_s^{\top}\boldsymbol{\theta}^*)\Big)}_{\text{exploitation}}$$

$$+ \underbrace{\sum_{\ell\in[L]}\sum_{s\in\Psi_{T+1,\ell}}\Big(2\mathbf{x}_s^{*\top}\boldsymbol{\theta}^* - (\mathbf{x}_s^{\top}\boldsymbol{\theta}^* + \mathbf{y}_s^{\top}\boldsymbol{\theta}^*)\Big)}_{\text{exploration}}\Bigg].$$

We bound the incurred regret of each part separately.

For any round $s \in T/\cup_{\ell\in[L]}\Psi_{T+1,\ell}$, the given condition for exploitation indicates the existence of a layer $\ell_s$ such that $\|\mathbf{x}_s - \mathbf{y}_s\|_{\widehat{\Sigma}_{s,\ell}^{-1}} \leq \alpha$ for all $\mathbf{x}_s, \mathbf{y}_s \in \mathcal{A}_{s,\ell}$. Using the Cauchy inequality and the MLE described in Section 4.2, we can show that the regret incurred in round $s$ is smaller than $3\widehat{\beta}_{s,\ell_s} \cdot \alpha$. Considering the simple upper bound $\widehat{\beta}_{s,\ell_s} \leq \widetilde{O}(\sqrt{T})$ and $\alpha = T^{-3/2}$, the regret for one exploitation round does not exceed $\widetilde{O}(1/T)$. Consequently, the cumulative regret is

$$\sum_{s\in[T]/(\cup_{\ell\in[L]}\Psi_{T+1,\ell})}\Big(2\mathbf{x}_s^{*\top}\boldsymbol{\theta}^* - (\mathbf{x}_s^{\top}\boldsymbol{\theta}^* + \mathbf{y}_s^{\top}\boldsymbol{\theta}^*)\Big) \leq \widetilde{O}(1)., \tag{5.1}$$

which is a low-order term in total regret.

In the exploration part, the regret is the cumulative regret encountered within each layer. We analyze the low layers and high layers distinctly. For $\ell \leq \ell^* = \Big\lceil \log_2\Big(64(L_\mu/\kappa_\mu)\sqrt{\log(4(T+1)^2 L/\delta)}\Big)\Big\rceil$, the incurred regret can be upper bounded by the number of rounds in this layer

$$\sum_{s\in\Psi_{T+1,\ell}}\big(2\mathbf{x}_s^{*\top}\boldsymbol{\theta}^* - (\mathbf{x}_s^{\top}\boldsymbol{\theta}^* + \mathbf{y}_s^{\top}\boldsymbol{\theta}^*)\big) \leq 4|\Psi_{T+1,\ell}|.$$

Moreover, $|\Psi_{T+1,\ell}|$ can be upper bounded by

$$|\Psi_{T+1,\ell}| \leq 2^{2\ell}d\log\big(1 + 2^{2\ell}AT/d\big) \leq O\Big(\frac{L_\mu^2}{\kappa_\mu^2}d\log\big(1 + 2^{2\ell^*}AT/d\big)\log\big(4(T+1)^2 L/\delta\big)\Big). \tag{5.2}$$

Thus the total regret for layers $\ell \leq \ell^*$ is bounded by $\widetilde{O}(d)$. For $\ell > \ell^*$, we can bound the cumulative regret incurred in each layer with

**Lemma 5.6.** With high probability, for all $\ell \in [L] \setminus \{1\}$, the regret incurred by the index set $\Psi_{T+1,\ell}$ is bounded by

$$\sum_{s\in\Psi_{T+1,\ell}}\Big(2\mathbf{x}_s^{*\top}\boldsymbol{\theta}^* - \big(\mathbf{x}_s^{\top}\boldsymbol{\theta}^* + \mathbf{y}_s^{\top}\boldsymbol{\theta}^*\big)\Big) \leq \widetilde{O}\Big(d \cdot 2^\ell\widehat{\beta}_{T,\ell-1}\Big).$$

By summing up the regret of all the layers, we can upper bound the total regret for layers $\ell > \ell^*$ as

$$\sum_{\ell\in[L]/[\ell^*]}\sum_{s\in\Psi_{T+1,\ell}}\Big(2\mathbf{x}_s^{*\top}\boldsymbol{\theta}^* - \big(\mathbf{x}_s^{\top}\boldsymbol{\theta}^* + \mathbf{y}_s^{\top}\boldsymbol{\theta}^*\big)\Big) \leq \widetilde{O}\Big(\frac{d}{\kappa_\mu}\sqrt{\sum_{t=1}^{T}\sigma_t^2} + \frac{d}{\kappa_\mu}\Big),$$

We can complete the proof of Theorem 5.1 by combining the regret in different parts together. For the detailed proof, please refer to Appendix E.

## 6 Experiments

**Experiment Setup.** We study the proposed algorithm in simulation to compare it with those that are also designed for contextual dueling bandits. Each experiment instance is simulated for $T = 4000$ rounds. The unknown parameter $\boldsymbol{\theta}^*$ to be estimated is generated at random and normalized to be a unit vector. The feature dimension is set to $d = 5$. A total of $|\mathcal{A}_t| = 2^d$ distinct contextual vectors are generated from $\{-1, 1\}^d$. In each round, given the arm pair selected by the algorithm, a response is generated according to the random process defined in Section 3. For each experiment, a total of 128

repeated runs are carried out. We tune the confidence radius of each algorithm to showcase the best performance. The average cumulative regret is reported in Figure 1 along with the standard deviation in the shaded region. The link function $\mu(\cdot)$ is set to be the logistic function. Our implementation is publicly available [1] .

**Algorithms.** We list the algorithms studied in this section as follows:

- `MaxInP`: Maximum Informative Pair by Saha (2021). It maintains an active set of possible optimal arms each round. The pairs are chosen on the basis of the maximum uncertainty in the difference between the two arms. Instead of using a warm-up period $\tau_0$ in their definition, we initialize $\mathbf{\Sigma}_0 = \lambda\mathbf{I}$ as regularization. When $\lambda = 0.001$ this approach empirically has no significant impact on regret performance compared to the warm-up method.

- `MaxPairUCB`: In this algorithm, we keep the MLE the same as `MaxInP`. However, we eliminate the need for an active set of arms, and the pair of arms that is picked is according to the term defined in (4.4).

- `CoLSTIM`: This method is from Bengs et al. (2022). First, they add randomly disturbed utilities to each arm and pick the arm that has the best estimation. They claim this step achieves better empirical performance. The second arm is chosen according to criteria as defined in (4.5).

- `VACDB`: The proposed variance-aware Algorithm 1 in this paper. $\alpha$ is set to this theoretical value according to Theorem 5.1. However, we note that for this specific experiment, $L = 4$ is enough to eliminate all suboptimal arms. The estimated $\widehat{\boldsymbol{\theta}}$ in one layer below is used to initialize the MLE of the upper layer when it is first reached to provide a rough estimate since the data is not shared among layers.

**Regret Comparison.** In Figure 1a we first notice that the proposed method `VACDB` has a better regret over other methods on average, demonstrating its efficiency. Second, the `MaxPairUCB` and `CoLSTIM` algorithm have a slight edge over the `MaxInP` algorithm empirically, which can be partially explained by the discussion in Section 4.4. The contributing factor for this could be that in `MaxInP` the chosen pair is solely based on uncertainty, while the other two methods choose at least one arm that maximizes the reward.

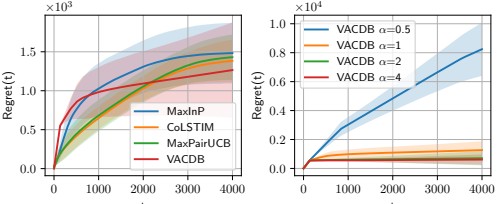

(a) Compare proposed algorithm with baselines.
(b) Variance-awareness of the proposed algorithm.

Figure 1: Experiments showing regret performance in various settings.

**Variance-Awareness.** In Figure 1b, we show the variance awareness of our algorithm by scaling the unknown parameter $\boldsymbol{\theta}^*$. Note that the variance of the Bernoulli distribution with parameter $p$ is $\sigma^2 = p(1-p)$. To generate high- and low-variance instances, we scale the parameter $\boldsymbol{\theta}^*$ by a ratio of $\alpha \in \{0.5, 1, 2, 4\}$. If $\alpha \geq 1$ then $p$ will be closer to $0$ or $1$ which results in a lower variance instance, and vice versa. In this plot, we show the result under four cases where the scale is set in an increasing manner, which corresponds to reducing the variance of each arm. With decreasing variance, our algorithm suffers less regret, which corresponds to the decrease in the $\sigma_t$ term in our main theorem.

## 7 CONCLUSION

We introduced a variance-aware method for contextual dueling bandits. An adaptive algorithm called `VACDB` is proposed. Theoretical analysis shows a regret upper bound depending on the observed variances in each round. The worst-case regret bound matches lower bound. Additionally, we conduct some simulated studies to show that the proposed algorithm reacts to instances with changing variance implied by the regret analysis. In the future, one of the possible directions is to consider a subset-wise comparison: In each round, a subset of size $K$ arms can be chosen from all arms, and the agent can only observe the best arm of the chosen subset. The dueling bandits model in this work can be treated as a special case of $K = 2$. Moreover, the preference probability is characterized by a generalized linear model, which may be a strong assumption for some real-world applications. We aim to generalize our results to broader nonlinear function classes, such as the function class with bounded Eluder dimension (Russo & Van Roy, 2013).

---

[1] https://github.com/uclaml/VACDB

ACKNOWLEDGEMENTS

We thank the anonymous reviewers and area chair for their helpful comments. QD, YW and QG are supported in part by the NSF grants CIF-1911168 and CPS-2312094. YW is also supported by UCLA Dissertation Year Fellowship. TJ and FF are supported in part by the NSF grant CIF-1908544. The views and conclusions contained in this paper are those of the authors and should not be interpreted as representing any funding agencies.

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

## A   COMPARISON WITH PRIOR WORKS

In this section, we provide a detailed discussion of the layered design, drawing a comparison with
`Sta'D` in Saha (2021) and `SupCoLSTIM` in Bengs et al. (2022). The general idea follows Auer
(2002), which focuses on maintaining a set of "high confidence promising arms". The algorithm
operates differently in two distinct scenarios. If there are some pairs $(\mathbf{x}_t, \mathbf{y}_t)$ in the current layer $\ell$
with high uncertainty, represented by $\|\mathbf{x}_t - \mathbf{y}_t\|_{\widehat{\mathbf{\Sigma}}_{t,\ell}^{-1}}$, we will explore those arm pairs. Conversely,
when achieving the desired accuracy, we eliminate suboptimal arms using our confidence set and
proceed to a subsequent layer demanding greater accuracy. This process continues until we reach
a sufficiently accurate high layer, at which we make decisions based on the remaining arms in the
confidence set and the estimated parameters $\widehat{\boldsymbol{\theta}}_{t,\ell}$.

In the final stage, `Sta'D` picks the first arm $\mathbf{x}_t$ as the one with the maximum estimated score,
followed by choosing its strongest challenger $\mathbf{y}_t$, which has the highest optimistic opportunity to
beat $\mathbf{x}_t$. `SupCoLSTIM` adopts a similar policy and distinguishes itself with a randomized learning
strategy by generating additive noise terms from an underlying perturbation distribution. Our arm
selection is based on the symmetric arm selection policy described in Section 4.4.

`Sta'D` and `SupCoLSTIM` choose the confidence set radius $\widehat{\beta}_{t,\ell}$ to be $2^{-\ell}$ in the $\ell$-th layer. In
comparison, our choice $\widehat{\beta}_{t,\ell}$ is defined in (4.3). As we mention in Section 4.3, apart from the
$2^{-\ell}$ dependency on the layer $\ell$, it also relies on the estimated variance. Such a variance-adaptive
confidence set radius helps to achieve the variance-aware regret bound.

## B   ADDITIONAL EXPERIMENT ON REAL-WORLD DATA

To showcase the performance of our algorithms in a real-world
setting, we use EventTime dataset (Zhang et al., 2016). In this
dataset, $K = 100$ historical events are compared in a pairwise
fashion by crowd-sourced workers. The data contains binary
response indicating which one of the events the worker thinks
precedes the other. There is no side information

$$\mathcal{A} = \{\mathbf{x}_i, i \in [K]\},$$

or the true parameter $\boldsymbol{\theta}^*$ readily available in the dataset. Thus,
we estimate them with pairwise comparison data. To achieve
this, let $C_{ij}, i, j \in [K]$ be the number of times event $j$ precedes
event $i$ labeled by the workers. The following MLE is used:

$$\underset{\{\mathbf{x}_i\}, \boldsymbol{\theta}}{\operatorname{argmax}} \sum_{i \in [K]} \sum_{j \in [K]} C_{ij} \log\left(\sigma((\mathbf{x}_i - \mathbf{x}_j)^\top \boldsymbol{\theta})\right).$$

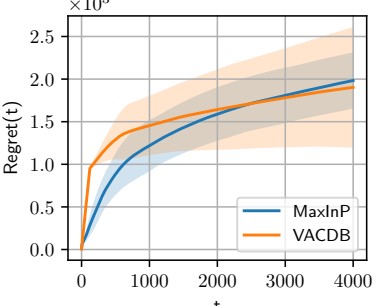

Figure 2: Regret comparison be-
tween `VACDB` and `MaxInP` on a
real-world dataset.

With the estimated $\mathcal{A}$ and $\boldsymbol{\theta}^*$, it is then possible to simulate the interactive process. We compared our
algorithm `VACDB` with `MaxInP` in Figure 2. We can see that after about 2500 rounds, our algorithm
starts to outperform `MaxInP` in terms of cumulative regret.

## C   DISCUSSION ON ARM SELECTION POLICIES

In this section, we present a detailed discussion for Section 4.4. We assume that in round $t$, we have
an estimator $\widehat{\boldsymbol{\theta}}_t$, a covariance matrix $\boldsymbol{\Sigma}_t = \lambda \mathbf{I} + \sum_{i=1}^{t-1}(\mathbf{x}_i - \mathbf{y}_i)(\mathbf{x}_i - \mathbf{y}_i)^\top$ and a concentration
inequality with confidence radius $\beta_t$,

$$\|\widehat{\boldsymbol{\theta}}_t - \boldsymbol{\theta}^*\|_{\boldsymbol{\Sigma}_t} \leq \beta_t. \tag{C.1}$$

The three arm selection methods can be described as follows:

**Method 1:**   Following Saha (2021), let $\mathcal{C}_t$ be

$$\mathcal{C}_t = \{\mathbf{x} \in \mathcal{A}_t \mid (\mathbf{x} - \mathbf{y})^\top \widehat{\boldsymbol{\theta}}_t + \beta_t \|\mathbf{x} - \mathbf{y}\|_{\boldsymbol{\Sigma}_t^{-1}} \geq 0, \forall \mathbf{y} \in \mathcal{A}_t\}.$$

Then $\mathbf{x}_t^* \in \mathcal{C}_t$ because for any $\mathbf{y} \in \mathcal{A}_t$

$$(\mathbf{x}_t^* - \mathbf{y})^\top \widehat{\boldsymbol{\theta}}_t + \beta_t \|\mathbf{x}_t^* - \mathbf{y}\|_{\boldsymbol{\Sigma}_t^{-1}} = (\mathbf{x}_t^* - \mathbf{y})^\top(\widehat{\boldsymbol{\theta}}_t - \boldsymbol{\theta}^*) + (\mathbf{x}_t^* - \mathbf{y})^\top \boldsymbol{\theta}^* + \beta_t \|\mathbf{x}_t^* - \mathbf{y}\|_{\boldsymbol{\Sigma}_t^{-1}}$$

$$\geq \beta_t \|\mathbf{x}_t^* - \mathbf{y}\|_{\Sigma_t^{-1}} - \|\mathbf{x}_t^* - \mathbf{y}\|_{\Sigma_t^{-1}}^\top \|\widehat{\boldsymbol{\theta}}_t - \boldsymbol{\theta}^*\|_{\boldsymbol{\Sigma}_t}$$

$$\geq 0,$$

where the first inequality holds due to Cauchy-Schwarz inequality and $\mathbf{x}_t^*$ is the optimal arm in round $t$. The second inequality holds due to (C.1).

The arms selected in round $t$ are $\mathbf{x}_t, \mathbf{y}_t = \mathrm{argmax}_{\mathbf{x}, \mathbf{y} \in \mathcal{C}_t} \|\mathbf{x} - \mathbf{y}\|_{\boldsymbol{\Sigma}_t^{-1}}$ Then the regret in round $t$ can be decomposed as

$$
\begin{aligned}
2r_t &= 2\mathbf{x}_t^{*\top}\boldsymbol{\theta}^* - (\mathbf{x}_t + \mathbf{y}_t)^\top \boldsymbol{\theta}^* \\
&= (\mathbf{x}_t^* - \mathbf{x}_t)^\top \boldsymbol{\theta}^* + (\mathbf{x}_t^* - \mathbf{y}_t)^\top \boldsymbol{\theta}^* \\
&= (\mathbf{x}_t^* - \mathbf{x}_t)^\top (\boldsymbol{\theta}^* - \widehat{\boldsymbol{\theta}}_t) + (\mathbf{x}_t^* - \mathbf{x}_t)^\top \widehat{\boldsymbol{\theta}}_t + (\mathbf{x}_t^* - \mathbf{y}_t)^\top (\boldsymbol{\theta}^* - \widehat{\boldsymbol{\theta}}_t) + (\mathbf{x}_t^* - \mathbf{y}_t)^\top \widehat{\boldsymbol{\theta}}_t \\
&\leq (\mathbf{x}_t^* - \mathbf{x}_t)^\top (\boldsymbol{\theta}^* - \widehat{\boldsymbol{\theta}}_t) + \beta_t \|\mathbf{x}_t^* - \mathbf{x}_t\|_{\boldsymbol{\Sigma}_t^{-1}} + (\mathbf{x}_t^* - \mathbf{y}_t)^\top (\boldsymbol{\theta}^* - \widehat{\boldsymbol{\theta}}_t) + \beta_t \|\mathbf{x}_t^* - \mathbf{y}_t\|_{\boldsymbol{\Sigma}_t^{-1}} \\
&\leq \|\mathbf{x}_t^* - \mathbf{x}_t\|_{\boldsymbol{\Sigma}_t^{-1}} \|\boldsymbol{\theta}^* - \widehat{\boldsymbol{\theta}}_t\|_{\boldsymbol{\Sigma}_t} + \beta_t \|\mathbf{x}_t^* - \mathbf{x}_t\|_{\boldsymbol{\Sigma}_t^{-1}} \\
&\quad + \|\mathbf{x}_t^* - \mathbf{y}_t\|_{\boldsymbol{\Sigma}_t^{-1}} \|\boldsymbol{\theta}^* - \widehat{\boldsymbol{\theta}}_t\|_{\boldsymbol{\Sigma}_t} + \beta_t \|\mathbf{x}_t^* - \mathbf{y}_t\|_{\boldsymbol{\Sigma}_t^{-1}} \\
&\leq 2\beta_t \|\mathbf{x}_t^* - \mathbf{x}_t\|_{\boldsymbol{\Sigma}_t^{-1}} + 2\beta_t \|\mathbf{x}_t^* - \mathbf{y}_t\|_{\boldsymbol{\Sigma}_t^{-1}} \\
&\leq 4\beta_t \|\mathbf{x}_t - \mathbf{y}_t\|_{\boldsymbol{\Sigma}_t^{-1}},
\end{aligned}
$$

where the first inequality holds because the choice $\mathbf{x}_t, \mathbf{y}_t \in \mathcal{C}_t$. The second inequality holds due to Cauchy-Schwarz inequality. The third inequality holds due to (C.1). The last inequality holds due to $\mathbf{x}_t^* \in \mathcal{C}_t$, $\mathbf{x}_t, \mathbf{y}_t = \mathrm{argmax}_{\mathbf{x}, \mathbf{y} \in \mathcal{C}_t} \|\mathbf{x} - \mathbf{y}\|_{\boldsymbol{\Sigma}_t^{-1}}$.

**Method 2:** Following Bengs et al. (2022), we choose the first arm as

$$
\mathbf{x}_t = \mathrm{argmax}_{\mathbf{x} \in \mathcal{A}_t} \mathbf{x}^\top \widehat{\boldsymbol{\theta}}_t.
$$

Then choose the second arm as

$$
\mathbf{y}_t = \mathrm{argmax}_{\mathbf{y} \in \mathcal{A}_t} \mathbf{y}^\top \widehat{\boldsymbol{\theta}}_t + 2\beta_t \|\mathbf{x}_t - \mathbf{y}\|_{\boldsymbol{\Sigma}_t^{-1}},
$$

The regret in round $t$ can be decomposed as

$$
\begin{aligned}
2r_t &= 2\mathbf{x}_t^{*\top}\boldsymbol{\theta}^* - (\mathbf{x}_t + \mathbf{y}_t)^\top \boldsymbol{\theta}^* \\
&= 2(\mathbf{x}_t^* - \mathbf{x}_t)^\top \boldsymbol{\theta}^* + (\mathbf{x}_t - \mathbf{y}_t)^\top \boldsymbol{\theta}^* \\
&= 2(\mathbf{x}_t^* - \mathbf{x}_t)^\top (\boldsymbol{\theta}^* - \widehat{\boldsymbol{\theta}}_t) + 2(\mathbf{x}_t^* - \mathbf{x}_t)^\top \widehat{\boldsymbol{\theta}}_t + (\mathbf{x}_t - \mathbf{y}_t)^\top (\boldsymbol{\theta}^* - \widehat{\boldsymbol{\theta}}_t) + (\mathbf{x}_t - \mathbf{y}_t)^\top \widehat{\boldsymbol{\theta}}_t \\
&\leq 2\|\mathbf{x}_t^* - \mathbf{x}_t\|_{\boldsymbol{\Sigma}_t^{-1}} \|\boldsymbol{\theta}^* - \widehat{\boldsymbol{\theta}}_t\|_{\boldsymbol{\Sigma}_t} + (\mathbf{x}_t^* - \mathbf{x}_t)^\top \widehat{\boldsymbol{\theta}}_t \\
&\quad + \|\mathbf{x}_t - \mathbf{y}_t\|_{\boldsymbol{\Sigma}_t^{-1}} \|\boldsymbol{\theta}^* - \widehat{\boldsymbol{\theta}}_t\|_{\boldsymbol{\Sigma}_t} + (\mathbf{x}_t - \mathbf{y}_t)^\top \widehat{\boldsymbol{\theta}}_t \\
&\leq 2\beta_t \|\mathbf{x}_t^* - \mathbf{x}_t\|_{\boldsymbol{\Sigma}_t^{-1}} + (\mathbf{x}_t^* - \mathbf{y}_t)^\top \widehat{\boldsymbol{\theta}}_t + \beta_t \|\mathbf{x}_t - \mathbf{y}_t\|_{\boldsymbol{\Sigma}_t^{-1}} \\
&\leq \mathbf{y}_t^\top \widehat{\boldsymbol{\theta}}_t + 2\beta_t \|\mathbf{x}_t - \mathbf{y}_t\|_{\boldsymbol{\Sigma}_t^{-1}} - \mathbf{x}_t^{*\top}\widehat{\boldsymbol{\theta}}_t + (\mathbf{x}_t^* - \mathbf{y}_t)^\top \widehat{\boldsymbol{\theta}}_t + \beta_t \|\mathbf{x}_t - \mathbf{y}_t\|_{\boldsymbol{\Sigma}_t^{-1}} \\
&= 3\beta_t \|\mathbf{x}_t - \mathbf{y}_t\|_{\boldsymbol{\Sigma}_t^{-1}},
\end{aligned}
$$

where the first inequality holds due to the Cauchy-Schwarz inequality and $\mathbf{x}_t^\top \widehat{\boldsymbol{\theta}}_t \geq \mathbf{x}_t^* \widehat{\boldsymbol{\theta}}_t$. The second inequality holds due to the Cauchy-Schwarz inequality. The third inequality holds due to $\mathbf{y}_t = \mathrm{argmax}_{\mathbf{y} \in \mathcal{A}_t} \mathbf{y}^\top \widehat{\boldsymbol{\theta}}_t + 2\beta_t \|\mathbf{x}_t - \mathbf{y}\|_{\Sigma_t^{-1}}$.

**Method 3:** In this method, we choose two arms as

$$
\mathbf{x}_t, \mathbf{y}_t = \mathrm{argmax}_{\mathbf{x}, \mathbf{y} \in \mathcal{A}_t} \left[ (\mathbf{x} + \mathbf{y})^\top \widehat{\boldsymbol{\theta}}_t + \beta_t \|\mathbf{x} - \mathbf{y}\|_{\widehat{\boldsymbol{\Sigma}}_t^{-1}} \right] \tag{C.2}
$$

Then the regret can be decomposed as

$$
\begin{aligned}
2r_t &= 2\mathbf{x}_t^{*\top}\boldsymbol{\theta}^* - (\mathbf{x}_t + \mathbf{y}_t)^\top \boldsymbol{\theta}^* \\
&= (\mathbf{x}_t^* - \mathbf{x}_t)^\top \boldsymbol{\theta}^* + (\mathbf{x}_t^* - \mathbf{y}_t)^\top \boldsymbol{\theta}^* \\
&= (\mathbf{x}_t^* - \mathbf{x}_t)^\top (\boldsymbol{\theta}^* - \widehat{\boldsymbol{\theta}}_t) + (\mathbf{x}_t^* - \mathbf{y}_t)^\top (\boldsymbol{\theta}^* - \widehat{\boldsymbol{\theta}}_t) + (2\mathbf{x}_t^* - \mathbf{x}_t - \mathbf{y}_t)^\top \widehat{\boldsymbol{\theta}}_t \\
&\leq \|\mathbf{x}_t^* - \mathbf{x}_t\|_{\boldsymbol{\Sigma}_t^{-1}} \|\boldsymbol{\theta}^* - \widehat{\boldsymbol{\theta}}_t\|_{\boldsymbol{\Sigma}_t} + \|\mathbf{x}_t^* - \mathbf{y}_t\|_{\boldsymbol{\Sigma}_t^{-1}} \|\boldsymbol{\theta}^* - \widehat{\boldsymbol{\theta}}_t\|_{\boldsymbol{\Sigma}_t} + (2\mathbf{x}_t^* - \mathbf{x}_t - \mathbf{y}_t)^\top \widehat{\boldsymbol{\theta}}_t \\
&\leq \beta_t \|\mathbf{x}_t^* - \mathbf{x}_t\|_{\boldsymbol{\Sigma}_t^{-1}} + \beta_t \|\mathbf{x}_t^* - \mathbf{y}_t\|_{\boldsymbol{\Sigma}_t^{-1}} + (2\mathbf{x}_t^* - \mathbf{x}_t - \mathbf{y}_t)^\top \widehat{\boldsymbol{\theta}}_t,
\end{aligned}
$$

where the first inequality holds due to the Cauchy-Schwarz inequality. The second inequality holds due to (C.1). Using (C.2), we have

$$(\mathbf{x}_t^* + \mathbf{x}_t)^\top \widehat{\boldsymbol{\theta}}_t + \beta_t \|\mathbf{x}_t^* - \mathbf{x}_t\|_{\widehat{\boldsymbol{\Sigma}}_t^{-1}} \leq (\mathbf{x}_t + \mathbf{y}_t)^\top \widehat{\boldsymbol{\theta}}_t + \beta_t \|\mathbf{x}_t - \mathbf{y}_t\|_{\widehat{\boldsymbol{\Sigma}}_t^{-1}}$$

$$(\mathbf{x}_t^* + \mathbf{y}_t)^\top \widehat{\boldsymbol{\theta}}_{t,\ell} + \beta_t \|\mathbf{x}_t^* - \mathbf{y}_t\|_{\widehat{\boldsymbol{\Sigma}}_t^{-1}} \leq (\mathbf{x}_t + \mathbf{y}_t)^\top \widehat{\boldsymbol{\theta}}_t + \beta_t \|\mathbf{x}_t - \mathbf{y}_t\|_{\widehat{\boldsymbol{\Sigma}}_t^{-1}}.$$

Adding the above two inequalities, we have

$$\beta_t \|\mathbf{x}_t^* - \mathbf{x}_t\|_{\boldsymbol{\Sigma}_t^{-1}} + \beta_t \|\mathbf{x}_t^* - \mathbf{y}_t\|_{\boldsymbol{\Sigma}_t^{-1}} \leq (\mathbf{x}_t + \mathbf{y}_t - 2\mathbf{x}_t^*)^\top \widehat{\boldsymbol{\theta}}_t + 2\beta_t \|\mathbf{x}_t - \mathbf{y}_t\|_{\widehat{\boldsymbol{\Sigma}}_t^{-1}}.$$

Therefore, we prove that the regret can be upper bounded by

$$2r_t \leq 2\beta_t \|\mathbf{x}_t - \mathbf{y}_t\|_{\widehat{\boldsymbol{\Sigma}}_t^{-1}}.$$

In conclusion, we can prove similar inequalities for the above three arm selection policies. To get an upper bound of regret, we can sum up the instantaneous regret in each round and use Lemma G.1 to obtain the final result.

## D A RIGOROUS PROOF FOR THE MLE

### D.1 DISCUSSION ON THE WEAKNESS

In the proof of Lemma E.1, for completeness, we need to prove that (4.1) has a unique solution. Following Li et al. (2017), we define a auxiliary function $G : \mathbb{R}^d \to \mathbb{R}^d$ as

$$G(\boldsymbol{\theta}) = \lambda \boldsymbol{\theta} + \sum_s w_s^2 \left[ \mu \left( (\mathbf{x}_s - \mathbf{y}_s)^\top \boldsymbol{\theta} \right) - \mu \left( (\mathbf{x}_s - \mathbf{y}_s)^\top \boldsymbol{\theta}^* \right) \right] (\mathbf{x}_s - \mathbf{y}_s).$$

Using the condition that the minimum eigenvalue of the covariance matrix is strictly positive, we can prove that $G$ is injective and $\widehat{\boldsymbol{\theta}}$ is the solution of (4.1) is equivalent to $G(\widehat{\boldsymbol{\theta}}) = Z$, where $Z$ is a quantity dependent on the stochastic noise. In Li et al. (2017), there is a minor weakness in asserting the existence and uniqueness of the solution with $\widehat{\boldsymbol{\theta}} = G^{-1}(Z)$, without confirming whether $Z$ lies in the range of $G$. We solve this problem with the classical Brouwer invariance of domain theorem in algebraic topology:

**Theorem D.1** (Brouwer 1911). Let $U$ be an open subset of $\mathbb{R}^d$, and let $f : U \to \mathbb{R}^d$ be a continuous injective map. Then $f(U)$ is also open.

We complete the proof by proving $G(\mathbb{R}^d)$ is both open and closed and therefore (4.1) has a unique solution.

### D.2 A DETAILED PROOF

We will prove that the function $G$ is a bijection from $\mathbb{R}^d$ to $\mathbb{R}^d$. We first show it's injective. The proof idea is similar to Theorem 1 in Li et al. (2017). With the mean value theorem, for any $\boldsymbol{\theta}_1, \boldsymbol{\theta}_2 \in \mathbb{R}^d$, there exists $m \in [0, 1]$ and $\bar{\boldsymbol{\theta}} = m\boldsymbol{\theta}_1 + (1 - m)\boldsymbol{\theta}_2$, such that the following equation holds,

$$G(\boldsymbol{\theta}_1) - G(\boldsymbol{\theta}_2)$$
$$= \lambda(\boldsymbol{\theta}_1 - \boldsymbol{\theta}_2) + \sum_s w_s^2 \left[ \mu \left( (\mathbf{x}_s - \mathbf{y}_s)^\top \boldsymbol{\theta}_1 \right) - \mu \left( (\mathbf{x}_s - \mathbf{y}_s)^\top \boldsymbol{\theta}_2 \right) \right] (\mathbf{x}_s - \mathbf{y}_s)$$
$$= \left[ \lambda \mathbf{I} + \sum_s w_s^2 \dot{\mu} \left( (\mathbf{x}_s - \mathbf{y}_s)^\top \bar{\boldsymbol{\theta}} \right) (\mathbf{x}_s - \mathbf{y}_s)(\mathbf{x}_s - \mathbf{y}_s)^\top \right] (\boldsymbol{\theta}_1 - \boldsymbol{\theta}_2).$$

We define $F(\bar{\boldsymbol{\theta}})$ as

$$F(\bar{\boldsymbol{\theta}}) = \left[ \lambda \mathbf{I} + \sum_s w_s^2 \dot{\mu} \left( (\mathbf{x}_s - \mathbf{y}_s)^\top \bar{\boldsymbol{\theta}} \right) (\mathbf{x}_s - \mathbf{y}_s)(\mathbf{x}_s - \mathbf{y}_s)^\top \right].$$

Using $\dot{\mu}(\cdot) \geq \kappa_\mu > 0$ and $\inf_s w_s^2 > 0$, we have $F(\bar{\boldsymbol{\theta}})$ is positive definite. Therefore, we prove that when $\boldsymbol{\theta}_1 \neq \boldsymbol{\theta}_2$, $G_{t,\ell}(\boldsymbol{\theta}_1) \neq G_{t,\ell}(\boldsymbol{\theta}_2)$. That is to say, $G_{t,\ell}$ is an injection from $\mathbb{R}^d$ to $\mathbb{R}^d$.

Next, we prove $G$ is surjective. The classical Brouwer invariance of domain theorem (Theorem G.4) in algebraic topology indicates that $G$ is an open map, and thus $G(\mathbb{R}^d)$ is an open set. On the other

hand, the minimum eigenvalue of $F(\bar{\boldsymbol{\theta}})$ is strictly positive. Therefore, $F(\bar{\boldsymbol{\theta}})$ is invertible, and we have

$$\boldsymbol{\theta}_1 - \boldsymbol{\theta}_2 = F(\bar{\boldsymbol{\theta}})^{-1} \left[ G_{t,\ell}(\boldsymbol{\theta}_1) - G_{t,\ell}(\boldsymbol{\theta}_2) \right]. \tag{D.1}$$

Let $\{G_{t,\ell}(\boldsymbol{\theta}_i)\}_{i=1}^{\infty}$ be a Cauchy sequence in $G(\mathbb{R}^d)$. Using (D.1) and the fact that $\lambda_{\min}(F(\bar{\boldsymbol{\theta}})) \geq \lambda > 0$, we have for any $m > n$,

$$\|\boldsymbol{\theta}_m - \boldsymbol{\theta}_n\|_2 \leq \frac{1}{\lambda} \|G(\boldsymbol{\theta}_m) - G(\boldsymbol{\theta}_n)\|_2.$$

This inequality shows that $\{\boldsymbol{\theta}_i\}_{i=1}^{\infty}$ is also a Cauchy sequence. With the completeness of the space $\mathbb{R}^d$, the limit $\lim_{i \to \infty} \boldsymbol{\theta}_i = \boldsymbol{\theta}$ exists. By the continuity of the function $G$, we have

$$\lim_{i \to \infty} G(\boldsymbol{\theta}_i) = G(\boldsymbol{\theta}) \in G(\mathbb{R}^d).$$

Therefore, $G(\mathbb{R}^d)$ is also closed. We have proved that $G(\mathbb{R}^d)$ is both open and closed. Using $\mathbb{R}^d$ is connected, we have proved that $G(\mathbb{R}^d) = \mathbb{R}^d$, i.e. $G_{t,\ell}$ is subjective.

In conclusion, the function $G$ is invertible, and (4.1) has a unique solution.

## E  PROOF OF THEOREM 5.1

In this section, we assume (4.1) has a unique solution $\widehat{\boldsymbol{\theta}}_{t+1,\ell}$, which is essential in our analysis. A detailed discussion is in Section D.

We first need the concentration inequality for the MLE.

**Lemma E.1.** With probability at least $1 - \delta$, the following concentration inequality holds for all round $t \geq 2$ and layer $\ell \in [L]$ simultaneously:

$$\left\| \widehat{\boldsymbol{\theta}}_{t,\ell} - \boldsymbol{\theta}^* \right\|_{\widehat{\boldsymbol{\Sigma}}_{t,\ell}} \leq \frac{2^{-\ell}}{\kappa_\mu} \left[ 16 \sqrt{\sum_{s \in \boldsymbol{\Psi}_{t,\ell}} w_s^2 \sigma_s^2 \log(4t^2 L/\delta)} + 6 \log(4t^2 L/\delta) \right] + 2^{-\ell}.$$

With this lemma, we have the following event holds with high probability:

$$\mathcal{E} = \left\{ \left\| \widehat{\boldsymbol{\theta}}_{t,\ell} - \boldsymbol{\theta}^* \right\|_{\widehat{\boldsymbol{\Sigma}}_{t,\ell}} \leq \frac{2^{-\ell}}{\kappa_\mu} \left[ 16 \sqrt{\sum_{s \in \boldsymbol{\Psi}_{t,\ell}} w_s^2 \sigma_s^2 \log(4t^2 L/\delta)} + 6 \log(4t^2 L/\delta) \right] + 2^{-\ell} \text{ for all } t, \ell \right\}.$$

Lemma E.1 shows that $\mathbb{P}[\mathcal{E}] \geq 1 - \delta$. For our choice of $\widehat{\beta}_{t,\ell}$ defined in (4.3), we define the following event:

$$\mathcal{E}^{\text{bonus}} = \left\{ \widehat{\beta}_{t,\ell} \geq \frac{2^{-\ell}}{\kappa} \left[ 16 \sqrt{\sum_{s \in \boldsymbol{\Psi}_{t,\ell}} w_s^2 \sigma_s^2 \log(4t^2 L/\delta)} + 6 \log(4t^2 L/\delta) \right] + 2^{-\ell}, \text{ for all } t, \ell \right\}.$$

The following two lemmas show that the event $\mathcal{E}_\ell^{\text{bonus}}$ holds with high probability.

**Lemma E.2.** With probability at least $1 - \delta$, for all $t \geq 2$, $\ell \in [L]$, the following two inequalties hold simultaneously.

$$\sum_{s \in \boldsymbol{\Psi}_{t,\ell}} w_s^2 \sigma_s^2 \leq 2 \sum_{s \in \boldsymbol{\Psi}_{t,\ell}} w_s^2 \epsilon_s^2 + \frac{14}{3} \log(4t^2 L/\delta).$$

$$\sum_{s \in \boldsymbol{\Psi}_{t,\ell}} w_s^2 \epsilon_s^2 \leq \frac{3}{2} \sum_{s \in \boldsymbol{\Psi}_{t,\ell}} w_s^2 \sigma_s^2 + \frac{7}{3} \log(4t^2 L/\delta).$$

**Lemma E.3.** Suppose that the inequalities in Lemma E.2 and the event $\mathcal{E}$ hold. For all $t \geq 2$ and $\ell \in [L]$ such that $2^\ell \geq 64(L_\mu/\kappa_\mu)\sqrt{\log(4(T+1)^2 L/\delta)}$, the following inequalities hold

$$\sum_{s \in \boldsymbol{\Psi}_{t,\ell}} w_s^2 \sigma_s^2 \leq 8 \sum_{s \in \boldsymbol{\Psi}_{t,\ell}} w_s^2 \left( o_s - \mu \left( (\mathbf{x}_s - \mathbf{y}_s)^\top \widehat{\boldsymbol{\theta}}_{t,\ell} \right) \right)^2 + 18 \log(4(t+1)^2 L/\delta).$$

$$\sum_{s \in \boldsymbol{\Psi}_{t,\ell}} w_s^2 \left( o_s - \mu \left( (\mathbf{x}_s - \mathbf{y}_s)^\top \widehat{\boldsymbol{\theta}}_{t,\ell} \right) \right)^2 \leq 4 \sum_{s \in \boldsymbol{\Psi}_{t,\ell}} w_s^2 \sigma_s^2 + 8 \log(4(t+1)^2 L/\delta).$$

Recall that with our choice of $\widehat{\beta}_{t,\ell}$ in (4.3), the inequality in $\mathcal{E}^{\text{bonus}}$ holds naturally when $2^\ell < 64(L_\mu/\kappa_\mu)\sqrt{\log(4(T+1)^2L/\delta)}$. Combining Lemma E.2, Lemma E.3 and $\mathbb{P}[\mathcal{E}] \geq 1 - \delta$, after taking a union bound, we have proved $\mathbb{P}[\mathcal{E}^{\text{bonus}} \cap \mathcal{E}] \geq 1 - 2\delta$.

**Lemma E.4.** Suppose the high probability events $\mathcal{E}^{\text{bonus}}$ and $\mathcal{E}$ holds. Then for all $t \geq 1$ and $\ell \in [L]$ such that the set $\mathcal{A}_{t,\ell}$ is defined, the contextual vector of the optimal arm $\mathbf{x}_t^*$ lies in $\mathcal{A}_{t,\ell}$.

Then we can bound the regret incurred in each layer separately.

**Lemma E.5.** Suppose the the high probability events $\mathcal{E}^{\text{bonus}}$ and $\mathcal{E}$ holds. Then for all $\ell \in [L]/1$, the regret incurred by the index set $\Psi_{T+1,\ell}$ is bounded by

$$\sum_{s \in \Psi_{T+1,\ell}} \left(2\mathbf{x}_s^{*\top}\boldsymbol{\theta}^* - (\mathbf{x}_s^\top\boldsymbol{\theta}^* + \mathbf{y}_s^\top\boldsymbol{\theta}^*)\right) \leq \widetilde{O}\left(d \cdot 2^\ell \widehat{\beta}_{T,\ell-1}\right).$$

With all these lemmas, we can prove Theorem 5.1.

*Proof of Theorem 5.1.* Conditioned on $\mathcal{E}^{\text{bonus}} \cap \mathcal{E}$, let

$$\ell^* = \left\lceil \log_2(64(L_\mu/\kappa_\mu)\sqrt{\log(4(T+1)^2L/\delta))}\right\rceil.$$

Using the high probability event $\mathcal{E}^{\text{bonus}}$, Lemma E.4 and Lemma E.5, for any $\ell > \ell^*$, we have

$$\sum_{s \in \Psi_{T+1,\ell}} \left(2\mathbf{x}_s^{*\top}\boldsymbol{\theta}^* - (\mathbf{x}_s^\top\boldsymbol{\theta}^* + \mathbf{y}_s^\top\boldsymbol{\theta}^*)\right)$$

$$\leq \widetilde{O}\left(d \cdot 2^\ell \widehat{\beta}_{T,\ell-1}\right)$$

$$\leq \widetilde{O}\left(\frac{d}{\kappa_\mu}\sqrt{\sum_{s \in \Psi_{T+1,\ell}} w_s^2\left(o_s - \mu((\mathbf{x}_s - \mathbf{y}_s)^\top\widehat{\boldsymbol{\theta}}_{T+1,\ell})\right)^2 + 1} + 1\right)$$

$$\leq \widetilde{O}\left(\frac{d}{\kappa_\mu}\sqrt{\sum_{t=1}^T \sigma_t^2} + \frac{d}{\kappa_\mu} + 1\right), \tag{E.1}$$

where the first inequality holds due to Lemma E.5. The second inequality holds due to the definition 4.3. The last inequality holds due to Lemma E.3 and $w_s \leq 1$.

For $\ell \in [\ell^*]$, we have

$$\sum_{s \in \Psi_{T+1,\ell}} \left(2\mathbf{x}_s^{*\top}\boldsymbol{\theta}^* - (\mathbf{x}_s^\top\boldsymbol{\theta}^* + \mathbf{y}_s^\top\boldsymbol{\theta}^*)\right)$$

$$\leq 4|\Psi_{T+1,\ell}|$$

$$= 2^{2\ell+2} \sum_{s \in \Psi_{T+1,\ell}} \|w_s(\mathbf{x}_s - \mathbf{y}_s)\|_{\widehat{\boldsymbol{\Sigma}}_{s,\ell}}^2$$

$$\leq 2^{2\ell+3}d\log(1 + T/(d\lambda))$$

$$= \widetilde{O}\left(\frac{dL_\mu^2}{\kappa_\mu^2}\right), \tag{E.2}$$

where the first equality holds due to our choice of $w_s$ such that $\|w_s(\mathbf{x}_s - \mathbf{y}_s)\|_{\widehat{\boldsymbol{\Sigma}}_{s,\ell}}^2$. The second inequality holds due to Lemma G.1. The last equality holds due to $\ell \leq \ell^*$

For any $s \in [T]/(\cup_{\ell \in [L]}\Psi_{T+1,\ell})$, we set $\ell_s$ as the value of layer such that $\|\mathbf{x}_s - \mathbf{y}_s\|_{\widehat{\boldsymbol{\Sigma}}_{s,\ell}^{-1}} \leq \alpha$ for all $\mathbf{x}_s, \mathbf{y}_s \in \mathcal{A}_{s,\ell}$ and then the while loop ends. By the choice of $\mathbf{x}_s, \mathbf{y}_s$ and $\mathbf{x}_s^* \in \mathcal{A}_{s,\ell_s}$ (Lemma E.4), we have

$$2\mathbf{x}_s^{*\top}\widehat{\boldsymbol{\theta}}_{s,\ell_s} \leq \mathbf{x}_s^\top\widehat{\boldsymbol{\theta}}_{s,\ell_s} + \mathbf{y}_s^\top\widehat{\boldsymbol{\theta}}_{s,\ell_s} + \widehat{\beta}_{s,\ell_s}\|\mathbf{x}_s - \mathbf{y}_s\|_{\widehat{\boldsymbol{\Sigma}}_{s,\ell_s}^{-1}}$$

$$\leq \mathbf{x}_s^\top\widehat{\boldsymbol{\theta}}_{s,\ell_s} + \mathbf{y}_s^\top\widehat{\boldsymbol{\theta}}_{s,\ell_s} + \widehat{\beta}_{s,\ell_s}\alpha, \tag{E.3}$$

where the last inequality holds because $\|\mathbf{x}_s - \mathbf{y}_s\|_{\widehat{\mathbf{\Sigma}}_{s,\ell}^{-1}} \le \alpha$ for all $\mathbf{x}_s, \mathbf{y}_s \in \mathcal{A}_{s,\ell}$. Then we have

$$
\sum_{s \in [T]/(\cup_{\ell \in [L]} \Psi_{T+1,\ell})} \left( 2\mathbf{x}_s^{*\top} \boldsymbol{\theta}^* - (\mathbf{x}_s^\top \boldsymbol{\theta}^* + \mathbf{y}_s^\top \boldsymbol{\theta}^*) \right)
$$

$$
= \sum_{s \in [T]/(\cup_{\ell \in [L]} \Psi_{T+1,\ell})} \left( 2\mathbf{x}_s^{*\top} \boldsymbol{\theta}^* - 2\mathbf{x}_s^{*\top} \widehat{\boldsymbol{\theta}}_{s,\ell_s} + \left( \mathbf{x}_s^\top \widehat{\boldsymbol{\theta}}_{s,\ell_s} - \mathbf{x}_s^\top \boldsymbol{\theta}^* \right) \right.
$$

$$
\left. + \left( \mathbf{y}_s^\top \widehat{\boldsymbol{\theta}}_{s,\ell_s} - \mathbf{y}_s^\top \boldsymbol{\theta}^* \right) + \left( 2\mathbf{x}_s^{*\top} \widehat{\boldsymbol{\theta}}_{s,\ell_s} - (\mathbf{x}_s^\top \widehat{\boldsymbol{\theta}}_{s,\ell_s} + \mathbf{y}_s^\top \widehat{\boldsymbol{\theta}}_{s,\ell_s}) \right) \right)
$$

$$
\le \sum_{s \in [T]/(\cup_{\ell \in [L]} \Psi_{T+1,\ell})} \left( \|\mathbf{x}_s^* - \mathbf{x}_s\|_{\widehat{\mathbf{\Sigma}}_{s,\ell_s}^{-1}} + \|\mathbf{x}_s^* - \mathbf{y}_s\|_{\widehat{\mathbf{\Sigma}}_{s,\ell_s}^{-1}} \right) \|\boldsymbol{\theta}^* - \widehat{\boldsymbol{\theta}}_{s,\ell_s}\|_{\widehat{\Sigma}_{s,\ell_s}} + \widehat{\beta}_{s,\ell_s} \alpha
$$

$$
\le \sum_{s \in [T]/(\cup_{\ell \in [L]} \Psi_{T+1,\ell})} 3\widehat{\beta}_{s,\ell_s} \alpha
$$

$$
\le T \cdot \widetilde{O}\left(1/T\right) = \widetilde{O}(1), \tag{E.4}
$$

where the first inequality holds due to the Cauchy-Schwarz inequality and (E.3). The third inequality holds due to $\|\mathbf{x}_s - \mathbf{y}_s\|_{\widehat{\mathbf{\Sigma}}_{s,\ell}^{-1}} \le \alpha$ for all $\mathbf{x}_s, \mathbf{y}_s \in \mathcal{A}_{s,\ell_s}$, $\mathbf{x}_s^* \in \mathcal{A}_{s,\ell_s}$ (Lemma E.4) and Lemma E.1. The third inequality holds due to our choice of $\widehat{\beta}_{s,\ell_s} \le \widetilde{O}(\sqrt{T})$ and $\alpha = 1/T^{3/2}$. Combining (E.1), (E.2), (E.4) together, we obtain

$$
\text{Regret}(T) = \widetilde{O}\left( \frac{d}{\kappa_\mu} \sqrt{\sum_{t=1}^T \sigma_t^2} + d\left(\frac{L_\mu^2}{\kappa_\mu^2} + \frac{1}{\kappa_\mu}\right) \right).
$$

$\square$

# F PROOF OF LEMMAS IN SECTION E

## F.1 PROOF OF LEMMA E.1

*Proof of Lemma E.1.* For a fixed $\ell \in [L]$, let $t \in \Psi_{T+1,\ell}$, $t \ge 2$, we define some auxiliary quantities:

$$
G_{t,\ell}(\boldsymbol{\theta}) = 2^{-2\ell} \kappa_\mu \boldsymbol{\theta} + \sum_{s \in \mathbf{\Psi}_{t,\ell}} w_s^2 \left[ \mu\left((\mathbf{x}_s - \mathbf{y}_s)^\top \boldsymbol{\theta}\right) - \mu\left((\mathbf{x}_s - \mathbf{y}_s)^\top \boldsymbol{\theta}^*\right) \right] (\mathbf{x}_s - \mathbf{y}_s)
$$

$$
\epsilon_t = o_t - \mu\left((\mathbf{x}_t - \mathbf{y}_t)^\top \boldsymbol{\theta}^*\right)
$$

$$
Z_{t,\ell} = \sum_{s \in \mathbf{\Psi}_{t,\ell}} w_s^2 \epsilon_s (\mathbf{x}_s - \mathbf{y}_s).
$$

Recall (4.1), $\widehat{\boldsymbol{\theta}}_{t,\ell}$ is the solution to

$$
2^{-2\ell} \kappa_\mu \widehat{\boldsymbol{\theta}}_{t,\ell} + \sum_{s \in \mathbf{\Psi}_{t,\ell}} w_s^2 \left( \mu\left((\mathbf{x}_s - \mathbf{y}_s)^\top \widehat{\boldsymbol{\theta}}_{t,\ell}\right) - o_s \right) (\mathbf{x}_s - \mathbf{y}_s) = \mathbf{0}. \tag{F.1}
$$

A simple transformation shows that (F.1) is equivalent to following equation,

$$
G_{t,\ell}\left(\widehat{\boldsymbol{\theta}}_{t,\ell}\right) = 2^{-2\ell} \kappa_\mu \widehat{\boldsymbol{\theta}}_{t,\ell} + \sum_{s \in \mathbf{\Psi}_{t,\ell}} w_s^2 \left[ \mu\left((\mathbf{x}_s - \mathbf{y}_s)^\top \widehat{\boldsymbol{\theta}}_{t,\ell}\right) - \mu\left((\mathbf{x}_s - \mathbf{y}_s)^\top \boldsymbol{\theta}^*\right) \right] (\mathbf{x}_s - \mathbf{y}_s)
$$

$$
= \sum_{s \in \mathbf{\Psi}_{t,\ell}} w_s^2 \left[ o_s - \mu\left((\mathbf{x}_s - \mathbf{y}_s)^\top \boldsymbol{\theta}^*\right) \right] (\mathbf{x}_s - \mathbf{y}_s)
$$

$$
= Z_{t,\ell}.
$$

We has proved $G_{t,\ell}$ is invertible in Section D and thus $\widehat{\boldsymbol{\theta}}_{t,\ell} = G_{t,\ell}^{-1}(Z_{t,\ell})$.

Moreover, we can see that $G_{t,\ell}(\boldsymbol{\theta}^*) = 2^{-2\ell}\kappa_\mu\boldsymbol{\theta}^*$. Recall $\widehat{\boldsymbol{\Sigma}}_{t,\ell} = 2^{-2\ell}\kappa_\mu\mathbf{I} + \sum_{s\in\boldsymbol{\Psi}_{t,\mu}} w_s^2(\mathbf{x}_s - \mathbf{y}_s)(\mathbf{x}_s - \mathbf{y}_s)^\top$. We have

$$\left\|G_{t,\ell}(\widehat{\boldsymbol{\theta}}_{t,\ell}) - G_{t,\ell}(\boldsymbol{\theta}^*)\right\|_{\widehat{\boldsymbol{\Sigma}}_{t,\ell}^{-1}}^2 = (\widehat{\boldsymbol{\theta}}_{t,\ell} - \boldsymbol{\theta}^*)^\top F(\bar{\boldsymbol{\theta}})\widehat{\boldsymbol{\Sigma}}_{t,\ell}^{-1}F(\bar{\boldsymbol{\theta}})(\widehat{\boldsymbol{\theta}}_{t,\ell} - \boldsymbol{\theta}^*)$$

$$\geq \kappa_\mu^2(\widehat{\boldsymbol{\theta}}_{t,\ell} - \boldsymbol{\theta}^*)^\top\widehat{\boldsymbol{\Sigma}}_{t,\ell}(\widehat{\boldsymbol{\theta}}_{t,\ell} - \boldsymbol{\theta}^*)$$

$$= \kappa_\mu^2\|\widehat{\boldsymbol{\theta}}_{t,\ell} - \boldsymbol{\theta}^*\|_{\widehat{\boldsymbol{\Sigma}}_{t,\ell}}^2,$$

where the first inequality holds because $\dot{\mu}(\cdot) \geq \kappa_\mu > 0$ and thus $F(\bar{\boldsymbol{\theta}}) \succeq \kappa_\mu\widehat{\boldsymbol{\Sigma}}_{t,\ell}$. Using the triangle inequality, we have

$$\left\|\widehat{\boldsymbol{\theta}}_{t,\ell} - \boldsymbol{\theta}^*\right\|_{\widehat{\boldsymbol{\Sigma}}_{t,\ell}} \leq 2^{-2\ell}\|\boldsymbol{\theta}^*\|_{\widehat{\boldsymbol{\Sigma}}_{t,\ell}^{-1}} + \frac{1}{\kappa_\mu}\|Z_{t,\ell}\|_{\widehat{\boldsymbol{\Sigma}}_{t,\ell}^{-1}}$$

$$\leq 2^{-\ell}\|\boldsymbol{\theta}^*\|_2 + \frac{1}{\kappa_\mu}\|Z_{t,\ell}\|_{\widehat{\boldsymbol{\Sigma}}_{t,\ell}^{-1}}.$$

To bound the $\|Z_{t,\ell}\|_{\widehat{\boldsymbol{\Sigma}}_{t,\ell}^{-1}}$ term, we use Lemma G.3. By the choice of $w_s$, for any $t \in \boldsymbol{\Psi}_{T+1,\ell}$, we have

$$\|w_t(\mathbf{x}_t - \mathbf{y}_t)\|_{\widehat{\boldsymbol{\Sigma}}_{t,\ell}^{-1}} = 2^{-\ell} \text{ and } w_t \leq 1.$$

We also have

$$\mathbb{E}[w_t^2\epsilon_t^2 \mid \mathcal{F}_t] \leq w_t^2\mathbb{E}[\epsilon_t^2 \mid \mathcal{F}_t] \leq w_t^2\sigma_t^2 \text{ and } |w_t\epsilon_t| \leq |\epsilon_t| \leq 1.$$

Therefore, Lemma G.3 shows that with probability at least $1 - \delta/L$, for all $t \in \boldsymbol{\Psi}_{T+1,\ell}$, the following inequality holds

$$\|Z_{t,\ell}\|_{\widehat{\boldsymbol{\Sigma}}_{t,\ell}^{-1}} \leq 16 \cdot 2^{-\ell}\sqrt{\sum_{s\in\boldsymbol{\Psi}_{t,\ell}} w_s^2\sigma_s^2\log(4t^2L/\delta)} + 6 \cdot 2^{-\ell}\log(4t^2L/\delta).$$

Finally, we get

$$\left\|\widehat{\boldsymbol{\theta}}_{t,\ell} - \boldsymbol{\theta}^*\right\|_{\widehat{\boldsymbol{\Sigma}}_{t,\ell}} \leq \frac{2^{-\ell}}{\kappa_\mu}\left[16\sqrt{\sum_{s\in\boldsymbol{\Psi}_{t,\ell}} w_s^2\sigma_s^2\log(4t^2L/\delta)} + 6\log(4t^2L/\delta)\right] + 2^{-\ell}.$$

Take a union bound on all $\ell \in [L]$, and then we finish the proof of Lemma E.1. $\qquad\square$

### F.2 PROOF OF LEMMA E.2

*Proof of Lemma E.2.* The proof of this lemma is similar to the proof of Lemma B.4 in Zhao et al. (2023a). For a fixed layer $\ell \in [L]$, using the definition of $\epsilon_s$ and $\sigma_s$, we have

$$\forall s \geq 1, \mathbb{E}[\epsilon_s^2 - \sigma_s^2|\mathbf{x}_{1:s}, \mathbf{y}_{1:s}, o_{1:s-1}] = 0.$$

Therefore, we have

$$\sum_{s\in\boldsymbol{\Psi}_{t,\ell}} \mathbb{E}[w_s^2(\epsilon_s^2 - \sigma_s^2)^2|\mathbf{x}_{1:s}, \mathbf{y}_{1:s}, o_{1:s-1}] \leq \sum_{s\in\boldsymbol{\Psi}_{t,\ell}} \mathbb{E}[w_s^2\epsilon_s^4|\mathbf{x}_{1:s}, \mathbf{y}_{1:s}, o_{1:s-1}]$$

$$\leq \sum_{s\in\boldsymbol{\Psi}_{t,\ell}} w_s^2\sigma_s^2,$$

where the last inequality holds due to the definition of $\sigma_s$ and $\epsilon_s \leq 1$. Then using Lemma G.2 and taking a union bound on all $\ell \in [L]$, for all $t \geq 2$, we have

$$\left|\sum_{s\in\boldsymbol{\Psi}_{t,\ell}} w_s^2(\epsilon_s^2 - \sigma_s^2)\right| \leq \sqrt{2\sum_{s\in\boldsymbol{\Psi}_{t,\ell}} w_s^2\sigma_s^2\log(4t^2L/\delta)} + \frac{2}{3}\cdot 2\log(4t^2L/\delta)$$

$$\leq \frac{1}{2}\sum_{s\in\boldsymbol{\Psi}_{t,\ell}} w_s^2\sigma_s^2 + \frac{7}{3}\log(4t^2L/\delta), \tag{F.2}$$

where we use the Young's inequality $ab \leq \frac{1}{2}a^2 + \frac{1}{2}b^2$. Finally, we finish the proof of Lemma E.2 by

$$
\begin{aligned}
\sum_{s \in \Psi_{t,\ell}} w_s^2 \sigma_s^2 &= \left| \sum_{s \in \Psi_{t,\ell}} w_s^2 \epsilon_s^2 - \sum_{s \in \Psi_{t,\ell}} w_s^2(\epsilon_s^2 - \sigma_s^2) \right| \\
&\leq \sum_{s \in \Psi_{t,\ell}} w_s^2 \epsilon_s^2 + \left| \sum_{s \in \Psi_{t,\ell}} w_s^2(\epsilon_s^2 - \sigma_s^2) \right| \\
&\leq \sum_{s \in \Psi_{t,\ell}} w_s^2 \epsilon_s^2 + \frac{1}{2} \sum_{s \in \Psi_{t,\ell}} w_s^2 \sigma_s^2 + \frac{7}{3} \log(4t^2 L/\delta),
\end{aligned} \tag{F.3}
$$

where the first inequality holds due to the triangle inequality. The second inequality holds due to (F.2). We also have

$$
\begin{aligned}
\sum_{s \in \Psi_{t,\ell}} w_s^2 \sigma_s^2 &= \left| \sum_{s \in \Psi_{t,\ell}} w_s^2 \epsilon_s^2 - \sum_{s \in \Psi_{t,\ell}} w_s^2(\epsilon_s^2 - \sigma_s^2) \right| \\
&\geq \sum_{s \in \Psi_{t,\ell}} w_s^2 \epsilon_s^2 - \left| \sum_{s \in \Psi_{t,\ell}} w_s^2(\epsilon_s^2 - \sigma_s^2) \right| \\
&\geq \sum_{s \in \Psi_{t,\ell}} w_s^2 \epsilon_s^2 - \frac{1}{2} \sum_{s \in \Psi_{t,\ell}} w_s^2 \sigma_s^2 - \frac{7}{3} \log(4t^2 L/\delta).
\end{aligned}
$$

The proof of this inequality is almost the same as (F.3). $\qquad \square$

### F.3 PROOF OF LEMMA E.3

*Proof of Lemma E.3.* For a fixed $\ell \in [L]$, Lemma E.2 indicates that

$$
\begin{aligned}
\sum_{s \in \Psi_{t,\ell}} w_s^2 \sigma_s^2 &\leq 2 \sum_{s \in \Psi_{t,\ell}} w_s^2 \epsilon_s^2 + \frac{14}{3} \log(4t^2 L/\delta) \\
&\leq \frac{14}{3} \log(4t^2 L/\delta) + 4 \sum_{s \in \Psi_{t,\ell}} w_s^2 \left( o_s - \mu\left((\mathbf{x}_s - \mathbf{y}_s)^\top \widehat{\boldsymbol{\theta}}_{t,\ell}\right) \right)^2 \\
&\quad + 4 \underbrace{\sum_{s \in \Psi_{t,\ell}} w_s^2 \left( \epsilon_s - \left( o_s - \mu\left((\mathbf{x}_s - \mathbf{y}_s)^\top \widehat{\boldsymbol{\theta}}_{t,\ell}\right) \right) \right)^2}_{(I)},
\end{aligned} \tag{F.4}
$$

where the second inequality holds due to the basic inequality $(a+b)^2 \leq 2a^2 + 2b^2$ for all $a, b \in \mathbb{R}$. Using our definition of $\epsilon_s$, $o_s = \mu\left((\mathbf{x}_s - \mathbf{y}_s)^\top \boldsymbol{\theta}^*\right) + \epsilon_s$. Thus, we have

$$
\begin{aligned}
(I) &= \sum_{s \in \Psi_{t,\ell}} w_s^2 \left( \epsilon_s - \left( o_s - \mu\left((\mathbf{x}_s - \mathbf{y}_s)^\top \widehat{\boldsymbol{\theta}}_{t,\ell}\right) \right) \right)^2 \\
&= \sum_{s \in \Psi_{t,\ell}} w_s^2 \left( \mu\left((\mathbf{x}_s - \mathbf{y}_s)^\top \widehat{\boldsymbol{\theta}}_{t,\ell}\right) - \mu\left((\mathbf{x}_s - \mathbf{y}_s)^\top \boldsymbol{\theta}^*\right) \right)^2 \\
&\leq L_\mu^2 \sum_{s \in \Psi_{t,\ell}} w_s^2 \left( (\mathbf{x}_s - \mathbf{y}_s)^\top \left( \widehat{\boldsymbol{\theta}}_{t,\ell} - \boldsymbol{\theta}^* \right) \right)^2,
\end{aligned} \tag{F.5}
$$

where the last inequality holds because the first order derivative of function $\mu$ is upper bounded by $L_\mu$ (Assumption 3.2). Moreover, by expanding the square, we have

$$
\begin{aligned}
(I) &\leq L_\mu^2 \sum_{s \in \Psi_{t,\ell}} w_s^2 \left( (\mathbf{x}_s - \mathbf{y}_s)^\top (\widehat{\boldsymbol{\theta}}_{t,\ell} - \boldsymbol{\theta}^*) \right)^2 \\
&= L_\mu^2 \sum_{s \in \Psi_{t,\ell}} (\widehat{\boldsymbol{\theta}}_{t,\ell} - \boldsymbol{\theta}^*)^\top w_s^2 (\mathbf{x}_s - \mathbf{y}_s)(\mathbf{x}_s - \mathbf{y}_s)^\top (\widehat{\boldsymbol{\theta}}_{t,\ell} - \boldsymbol{\theta}^*) \\
&= L_\mu^2 (\widehat{\boldsymbol{\theta}}_{t,\ell} - \boldsymbol{\theta}^*)^\top \left( \sum_{s \in \Psi_{t,\ell}} w_s^2 (\mathbf{x}_s - \mathbf{y}_s)(\mathbf{x}_s - \mathbf{y}_s)^\top \right) (\widehat{\boldsymbol{\theta}}_{t,\ell} - \boldsymbol{\theta}^*) \\
&\leq L_\mu^2 \left\| \widehat{\boldsymbol{\theta}}_{t,\ell} - \boldsymbol{\theta}^* \right\|_{\widehat{\boldsymbol{\Sigma}}_{t,\ell}}^2,
\end{aligned}
\tag{F.6}
$$

where the last inequality holds due to

$$
\widehat{\boldsymbol{\Sigma}}_{t,\ell} = 2^{-2\ell} \kappa_\mu \mathbf{I} + \sum_{s \in \Psi_{t,\ell}} w_s^2 (\mathbf{x}_s - \mathbf{y}_s)(\mathbf{x}_s - \mathbf{y}_s)^\top \succeq \sum_{s \in \Psi_{t,\ell}} w_s^2 (\mathbf{x}_s - \mathbf{y}_s)(\mathbf{x}_s - \mathbf{y}_s)^\top.
$$

Combining (F.5), (F.6) and the event $\mathcal{E}$ (Lemma E.1), we have

$$
\begin{aligned}
(I) &\leq \frac{2^{-2\ell} L_\mu^2}{\kappa_\mu^2} \left[ 16 \sqrt{\sum_{s \in \boldsymbol{\Psi}_{t,\ell}} w_s^2 \sigma_s^2 \log(4(t+1)^2 L/\delta)} + 6 \log(4(t+1)^2 L/\delta) + \kappa_\mu \right]^2 \\
&\leq \frac{2^{-2\ell} L_\mu^2}{\kappa_\mu^2} \left[ 512 \log(4(t+1)^2 L/\delta) \cdot \sum_{s \in \boldsymbol{\Psi}_{t,\ell}} w_s^2 \sigma_s^2 + 2 \left( 6 \log(4(t+1)^2 L/\delta) + \kappa_\mu \right)^2 \right],
\end{aligned}
$$

where the last inequality holds due to the basic inequality $(a+b)^2 \leq 2a^2 + 2b^2$ for all $a, b \in \mathbb{R}$. When $2^\ell \geq 64(L_\mu/\kappa_\mu)\sqrt{\log(4(t+1)^2 L/\delta)}$, we can further bound the above inequality by

$$
(I) \leq \frac{1}{8} \sum_{s \in \boldsymbol{\Psi}_{t+1,\ell}} w_s^2 \sigma_s^2 + \log(4(t+1)^2 L/\delta).
\tag{F.7}
$$

Subitituting (F.7) into (F.4), we have

$$
\begin{aligned}
\sum_{s \in \Psi_{t,\ell}} w_s^2 \sigma_s^2 \leq\ &4 \sum_{s \in \Psi_{t,\ell}} w_s^2 \left( o_s - \mu \left( (\mathbf{x}_s - \mathbf{y}_s)^\top \widehat{\boldsymbol{\theta}}_{t,\ell} \right) \right)^2 \\
&+ 9 \log(4(t+1)^2 L/\delta) + \frac{1}{2} \sum_{s \in \boldsymbol{\Psi}_{t,\ell}} w_s^2 \sigma_s^2.
\end{aligned}
$$

Therefore, we prove the first inequality in Lemma E.3 as follows

$$
\begin{aligned}
\sum_{s \in \Psi_{t,\ell}} w_s^2 \sigma_s^2 \leq\ &8 \sum_{s \in \Psi_{t,\ell}} w_s^2 \left( o_s - \mu \left( (\mathbf{x}_s - \mathbf{y}_s)^\top \widehat{\boldsymbol{\theta}}_{t,\ell} \right) \right)^2 \\
&+ 18 \log(4(t+1)^2 L/\delta).
\end{aligned}
$$

For the second inequality, we have

$$
\begin{aligned}
\sum_{s \in \Psi_{t,\ell}} &w_s^2 \left( o_s - \mu \left( (\mathbf{x}_s - \mathbf{y}_s)^\top \widehat{\boldsymbol{\theta}}_{t,\ell} \right) \right)^2 \\
&\leq 2 \sum_{s \in \Psi_{t,\ell}} w_s^2 \epsilon_s^2 + 2 \underbrace{\sum_{s \in \Psi_{t,\ell}} w_s^2 \left( \epsilon_s - \left( o_s - \mu \left( (\mathbf{x}_s - \mathbf{y}_s)^\top \widehat{\boldsymbol{\theta}}_{t,\ell} \right) \right) \right)^2}_{(I)}.
\end{aligned}
$$

We complete the proof of Lemma E.3.

$$\sum_{s \in \Psi_{t,\ell}} w_s^2 \left( o_s - \mu \left( (\mathbf{x}_s - \mathbf{y}_s)^\top \widehat{\boldsymbol{\theta}}_{t,\ell} \right) \right)^2$$

$$\leq 2 \sum_{s \in \Psi_{t,\ell}} w_s^2 \epsilon_s^2 + \frac{1}{4} \sum_{s \in \boldsymbol{\Psi}_{t,\ell}} w_s^2 \sigma_s^2 + 2 \log(4(t+1)^2 L/\delta)$$

$$\leq 2 \left( \frac{3}{2} \sum_{s \in \Psi_{t,\ell}} w_s^2 \sigma_s^2 + \frac{7}{3} \log(4t^2 L/\delta) \right) + \frac{1}{4} \sum_{s \in \boldsymbol{\Psi}_{t,\ell}} w_s^2 \sigma_s^2 + 2 \log(4(t+1)^2 L/\delta)$$

$$\leq 4 \sum_{s \in \Psi_{t,\ell}} w_s^2 \sigma_s^2 + 8 \log(4(t+1)^2 L/\delta),$$

where the first inequality holds due to (F.7). The second inequality holds due to Lemma E.2. $\qquad\square$

### F.4 PROOF OF LEMMA E.4

*Proof of Lemma E.4.* We prove it by induction. For $\ell = 1$, we initialze the set $\mathcal{A}_{t,1}$ to be $\mathcal{A}_t$, thus trivially $\mathbf{x}_t^* \in \mathcal{A}_{t,1}$. Now we suppose $\mathcal{A}_{t,\ell}$ is defined and $\mathbf{x}_t^* \in \mathcal{A}_{t,\ell}$. By the way $\mathcal{A}_{t,\ell+1}$ is constructed, $\mathcal{A}_{t,\ell+1}$ is defined only when $\|\mathbf{x} - \mathbf{y}\|_{\widehat{\boldsymbol{\Sigma}}_{t,\ell}^{-1}} \leq 2^{-\ell}$ for all $\mathbf{x}, \mathbf{y} \in \mathcal{A}_{t,\ell}$.

Let $\mathbf{x}_{\max} = \operatorname{argmax}_{\mathbf{x} \in \mathcal{A}_{t,\ell}} \mathbf{x}^\top \widehat{\boldsymbol{\theta}}_{t,\ell}$. Then we have

$$\mathbf{x}_t^{*\top} \widehat{\boldsymbol{\theta}}_{t,\ell} - \mathbf{x}_{\max}^\top \widehat{\boldsymbol{\theta}}_{t,\ell} = (\mathbf{x}_t^{*\top} \boldsymbol{\theta}^* - \mathbf{x}_{\max}^\top \boldsymbol{\theta}^*) + (\mathbf{x}_t^* - \mathbf{x}_{\max})^\top (\widehat{\boldsymbol{\theta}}_{t,\ell} - \boldsymbol{\theta}^*)$$

$$\geq -\|\mathbf{x}_t^* - \mathbf{x}_{\max}\|_{\widehat{\boldsymbol{\Sigma}}_{t,\ell}^{-1}} \cdot \|\widehat{\boldsymbol{\theta}}_{t,\ell} - \boldsymbol{\theta}^*\|_{\widehat{\boldsymbol{\Sigma}}_{t,\ell}},$$

where the inequality holds due to the Cauchy-Schwarz inequality and the fact $\mathbf{x}_t^* = \operatorname{argmax}_{\mathbf{x} \in \mathcal{A}_t} \mathbf{x}^\top \boldsymbol{\theta}^*$. With the inductive hypothesis, we know $\mathbf{x}_t^* \in \mathcal{A}_{t,\ell}$. Thus we have $\|\mathbf{x}_t^* - \mathbf{x}_{\max}\|_{\widehat{\boldsymbol{\Sigma}}_{t,\ell}^{-1}} \leq 2^{-\ell}$. Finally, with the inequality in Lemma E.1, we have

$$\mathbf{x}_t^{*\top} \widehat{\boldsymbol{\theta}}_{t,\ell} \geq \max_{\mathbf{x} \in \mathcal{A}_{t,\ell}} \mathbf{x}^\top \widehat{\boldsymbol{\theta}}_{t,\ell} - 2^{-\ell} \widehat{\beta}_{t,\ell}.$$

Therefore, we have $\mathbf{x}_t^* \in \mathcal{A}_{t,\ell+1}$, and we complete the proof of Lemma E.4 by induction. $\qquad\square$

### F.5 PROOF OF LEMMA E.5

*Proof of Lemma E.5.* For any $s \in \Psi_{T+1,\ell}$, due to the definition of $\Psi_{T+1,\ell}$ and our choice of $\mathbf{x}_s, \mathbf{y}_s$ (Algorithm 1 Line 14-16), we have $\mathbf{x}_s, \mathbf{y}_s \in \mathcal{A}_{s,\ell}$. Additionally, because the set $\mathcal{A}_{s,\ell}$ is defined, $\|\mathbf{x} - \mathbf{y}\|_{\widehat{\boldsymbol{\Sigma}}_{s,\ell-1}^{-1}} \leq 2^{-\ell+1}$ for all $\mathbf{x}, \mathbf{y} \in \mathcal{A}_{s,\ell-1}$. From Lemma E.4, we can see that $\mathbf{x}_s^* \in \mathcal{A}_{s,\ell}$. Combining these results, we have

$$\|\mathbf{x}_s^* - \mathbf{x}_s\|_{\widehat{\boldsymbol{\Sigma}}_{s,\ell-1}^{-1}} \leq 2^{-\ell+1}, \|\mathbf{x}_s^* - \mathbf{y}_s\|_{\widehat{\boldsymbol{\Sigma}}_{s,\ell-1}^{-1}} \leq 2^{-\ell+1}, \tag{F.8}$$

where we use the inclusion property $\mathcal{A}_{s,\ell} \subseteq \mathcal{A}_{s,\ell-1}$. Moreover, $\mathbf{x}_s, \mathbf{x}_s^* \in \mathcal{A}_{s,\ell}$ shows that

$$\mathbf{x}_s^\top \widehat{\boldsymbol{\theta}}_{s,\ell-1} \geq \max_{\mathbf{x} \in \mathcal{A}_{s,\ell-1}} \mathbf{x}^\top \widehat{\boldsymbol{\theta}}_{s,\ell-1} - 2^{-\ell+1} \widehat{\beta}_{s,\ell-1}$$

$$\geq \mathbf{x}_s^{*\top} \widehat{\boldsymbol{\theta}}_{s,\ell-1} - 2^{-\ell+1} \widehat{\beta}_{s,\ell-1}, \tag{F.9}$$

where we use $\mathbf{x}_s \in \mathcal{A}_{s,\ell-1}$. Similarly, we have

$$\mathbf{y}_s^\top \widehat{\boldsymbol{\theta}}_{s,\ell-1} \geq \mathbf{x}_s^{*\top} \widehat{\boldsymbol{\theta}}_{s,\ell-1} - 2^{-\ell+1} \widehat{\beta}_{s,\ell-1}. \tag{F.10}$$

Now we compute the regret incurred in round $s$.

$$
\begin{aligned}
2\mathbf{x}_s^{*\top}\boldsymbol{\theta}^* - \left(\mathbf{x}_s^\top\boldsymbol{\theta}^* + \mathbf{y}_s^\top\boldsymbol{\theta}^*\right) &= (\mathbf{x}_s^* - \mathbf{x}_s)^\top\boldsymbol{\theta}^* + (\mathbf{x}_s^* - \mathbf{y}_s)^\top\boldsymbol{\theta}^* \\
&\leq (\mathbf{x}_s^* - \mathbf{x}_s)^\top\widehat{\boldsymbol{\theta}}_{s,\ell-1} + \left|(\mathbf{x}_s^* - \mathbf{x}_s)^\top\left(\widehat{\boldsymbol{\theta}}_{s,\ell-1} - \boldsymbol{\theta}^*\right)\right| \\
&\quad + (\mathbf{x}_s^* - \mathbf{y}_s)^\top\widehat{\boldsymbol{\theta}}_{s,\ell-1} + \left|(\mathbf{x}_s^* - \mathbf{y}_s)^\top\left(\widehat{\boldsymbol{\theta}}_{s,\ell-1} - \boldsymbol{\theta}^*\right)\right| \\
&\leq 2^{-\ell+1}\widehat{\beta}_{s,\ell-1} + \|\mathbf{x}_s^* - \mathbf{x}_s\|_{\widehat{\boldsymbol{\Sigma}}_{s,\ell-1}^{-1}} \left\|\widehat{\boldsymbol{\theta}}_{s,\ell-1} - \boldsymbol{\theta}^*\right\|_{\widehat{\boldsymbol{\Sigma}}_{s,\ell-1}} \\
&\quad + 2^{-\ell+1}\widehat{\beta}_{s,\ell-1} + \|\mathbf{x}_s^* - \mathbf{y}_s\|_{\widehat{\boldsymbol{\Sigma}}_{s,\ell-1}^{-1}} \left\|\widehat{\boldsymbol{\theta}}_{s,\ell-1} - \boldsymbol{\theta}^*\right\|_{\widehat{\boldsymbol{\Sigma}}_{s,\ell-1}} \\
&\leq 8 \cdot 2^{-\ell}\widehat{\beta}_{s,\ell-1}, \tag{F.11}
\end{aligned}
$$

where the first inequality holds due to the basic inequality $x \leq |x|$ for all $x \in \mathbb{R}$. The second inequality holds due t (F.9), (F.10) and the Cauchy-Schwarz inequality. The last inequality holds due to (F.8) and Lemma E.1. Now we can return to the summation of regret on the index set $\Psi_{T+1,\ell}$.

$$
\begin{aligned}
\sum_{s\in\Psi_{T+1,\ell}} \left(2\mathbf{x}_s^{*\top}\boldsymbol{\theta}^* - (\mathbf{x}_s^\top\boldsymbol{\theta}^* + \mathbf{y}_s^\top\boldsymbol{\theta}^*)\right) &\leq \sum_{s\in\Psi_{T+1,\ell}} 8 \cdot 2^{-\ell}\widehat{\beta}_{s,\ell-1} \\
&\leq 8 \cdot 2^{-\ell}\widehat{\beta}_{T,\ell-1}|\Psi_{T+1,\ell}| \\
&\leq 8 \cdot 2^{\ell}\widehat{\beta}_{T,\ell-1} \sum_{s\in\Psi_{T+1,\ell}} \|\omega_s \cdot (\mathbf{x}_s - \mathbf{y}_s)\|_{\widehat{\boldsymbol{\Sigma}}_{s,\ell}^{-1}}^2 \\
&\leq 8 \cdot 2^{\ell}\widehat{\beta}_{T,\ell-1} \cdot 2d\log\left(1 + 2^{2\ell+2}T/d\right),
\end{aligned}
$$

where the first inequality holds due to (F.11). The second inequality holds due to our choice of $\omega_s$ such that $\|\omega_s \cdot (\mathbf{x}_s - \mathbf{y}_s)\|_{\widehat{\boldsymbol{\Sigma}}_{s,\ell}^{-1}} = 2^{-\ell}$. The last inequality holds due to Lemma G.1. Therefore, we complete the proof of Lemma E.5. $\qquad\square$

## G  AUXILIARY LEMMAS

**Lemma G.1** (Lemma 11, Abbasi-Yadkori et al. 2011)**.** For any $\lambda > 0$ and sequence $\{\mathbf{x}_k\}_{k=1}^K \subseteq \mathbb{R}^d$ for $k \in [K]$, define $\mathbf{Z}_k = \lambda\mathbf{I} + \sum_{i=1}^{k-1}\mathbf{x}_i\mathbf{x}_i^\top$. Then, provided that $\|\mathbf{x}_k\|_2 \leq L$ holds for all $k \in [K]$, we have

$$
\sum_{k=1}^K \min\{1, \|\mathbf{x}_k\|_{\mathbf{Z}_k^{-1}}^2\} \leq 2d\log(1 + KL^2/(d\lambda)).
$$

**Lemma G.2** (Freedman 1975)**.** Let $M, v > 0$ be fixed constants. Let $\{x_i\}_{i=1}^n$ be a stochastic process, $\{\mathcal{G}_i\}_{i\in[n]}$ be a filtration so that for all $i \in [n]$, $x_i$ is $\mathcal{G}_i$-measurable, while almost surely

$$
\mathbb{E}[x_i|\mathcal{G}_{i-1}] = 0, |x_i| \leq M, \sum_{i=1}^n \mathbb{E}[x_i^2|\mathcal{G}_{i-1}] \leq v.
$$

Then for any $\delta > 0$, with probability at least $1 - \delta$, we have

$$
\sum_{i=1}^n x_i \leq \sqrt{2v\log(1/\delta)} + 2/3 \cdot M\log(1/\delta).
$$

**Lemma G.3** (Zhao et al. 2023a)**.** Let $\{\mathcal{G}_k\}_{k=1}^\infty$ be a filtration, and $\{\mathbf{x}_k, \eta_k\}_{k\geq 1}$ be a stochastic process such that $\mathbf{x}_k \in \mathbb{R}^d$ is $\mathcal{G}_k$-measurable and $\eta_k \in \mathbb{R}$ is $\mathcal{G}_{k+1}$-measurable. Let $L, \sigma, \lambda, \epsilon > 0$, $\boldsymbol{\mu}^* \in \mathbb{R}^d$. For $k \geq 1$, let $y_k = \langle\boldsymbol{\mu}^*, \mathbf{x}_k\rangle + \eta_k$, where $\eta_k, \mathbf{x}_k$ satisfy

$$
\mathbb{E}[\eta_k \mid \mathcal{G}_k] = 0, |\eta_k| \leq R, \sum_{i=1}^k \mathbb{E}[\eta_i^2 \mid \mathcal{G}_i] \leq v_k, \text{ for } \forall k \geq 1.
$$

For $k \geq 1$, let $\mathbf{Z}_k = \lambda\mathbf{I} + \sum_{i=1}^k \mathbf{x}_i\mathbf{x}_i^\top$, $\mathbf{b}_k = \sum_{i=1}^k y_i\mathbf{x}_i$, $\boldsymbol{\mu}_k = \mathbf{Z}_k^{-1}\mathbf{b}_k$ and

$$
\beta_k = 16\rho\sqrt{v_k\log(4k^2/\delta)} + 6\rho R\log(4k^2/\delta),
$$

where $\rho \geq \sup_{k \geq 1} \|\mathbf{x}_k\|_{\mathbf{Z}_{k-1}^{-1}}$. Then, for any $0 < \delta < 1$, we have with probability at least $1 - \delta$,

$$\forall k \geq 1, \|\sum_{i=1}^{k} \mathbf{x}_i \eta_i\|_{\mathbf{Z}_k^{-1}} \leq \beta_k, \|\boldsymbol{\mu}_k - \boldsymbol{\mu}^*\|_{\mathbf{Z}_k} \leq \beta_k + \sqrt{\lambda}\|\boldsymbol{\mu}^*\|_2$$

**Theorem G.4** (Brouwer invariance of domain theorem,Brouwer 1911). Let $U$ be an open subset of $\mathbb{R}^d$, and let $f : U \rightarrow \mathbb{R}^d$ be a continuous injective map. Then $f(U)$ is also open.

