# OpenReview forum: "Variance-aware Regret Bounds for Stochastic Contextual Dueling Bandits"
_ICLR.cc/2024/Conference — ICLR 2024 poster_

### Official Review · Reviewer_83pH · 2023-10-15

**Soundness:** 3 good
**Presentation:** 3 good
**Contribution:** 2 fair
**Rating:** 5
**Confidence:** 3

**Summary:**

The paper addresses the problem of dueling bandits with a variance-aware regret bound. It introduces an algorithm for contextual dueling bandits, which accounts for uncertainty in pairwise comparisons between arms and provides a variance-aware regret bound. The paper highlights the importance of considering uncertainty in decision-making scenarios and proposes an efficient algorithm with a regret bound that depends on the variance of comparisons, dimensionality of context vectors, and the time horizon. The authors demonstrate the effectiveness of their method through empirical experiments.

**Strengths:**

The paper is well written and easy to follow.

The proposed algorithm is the first algorithm with a variance-aware regret bound. It has some novelty, and the symmetric arm selection is new.

The authors also support their theoretical claims with empirical experiments on synthetic data.

**Weaknesses:**

I feel this paper is an extension to [1], with similar algorithm proposed and regret analysis. The main algorithm has identical structure with the SAVE proposed in [1] based on the SupLin methodology, and it is not new and novel for the variance-aware contextual bandit problem. Under the generalized linear contextual dueling bandit setting used in the paper, we can regard $x_t - y_t$ as the contextual information, which would then be degeneralized to the ordinary generalized linear contextual bandit problem. I think the high similarity with the existing literature is the major weakness of this work.

Typos and minor issues:
1. title of subsection 4.3 (sysmmetric -> symmetric)
2. I may overlook. What is $\kappa$ in line 20 Algorithm 1 and Eqn. (4.3)?

[1] Variance-dependent regret bounds for linear bandits and reinforcement learning: adaptivity and computational efficiency. Zhao et al., COLT.

**Questions:**

In addition to my concerns in the above Weaknesses section,

1. Is the variance of the noise $\epsilon_t$ equal to $p_t(1-p_t)$ under the Bernoulli setting? In that case I feel the variance would be fully dependent on the arms selected in each round.

---

> ### Author Response · Authors · 2023-11-21
> **Response to Reviewer 83pH**
>
> Thank you for your valuable feedback. We address your questions point-by-point as follows.
>
> **Q1**: Comparison with SAVE in [1] and algorithms for generalized linear bandit problem.
>
> **A1**: Compared with SAVE, the main difference is that we consider the problem in the generalized linear model and dueling bandit setting, which has not been studied in the literature before. Note that SAVE is for linear bandits, rather than generalized linear (dueling) bandits. The nonlinearity of the problem requires us to consider MLE instead of linear regression. Since the linear regression problem has a closed-form solution, the analysis of MLE is more difficult.
>
> Compared with the generalized linear bandit, there is a key difference. You cannot simply regard $\mathbf{x} _ t- \mathbf y _ t$ as the feature vector to reduce dueling bandits to multi-armed bandits. Note that the regret in dueling bandits is defined as $\text{Regret}$ $(T) = \sum _ {t} $ $(\mathbf x _ t+ \mathbf y _ t)^\top \boldsymbol{\theta}^*$ $- 2 \mathbf x ^ {*\top} \boldsymbol{\theta} ^ *$. In this definition, the regret depends on $\mathbf x _ t+ \mathbf y _ t$ rather than $\mathbf x _ t- \mathbf y _ t$, i.e., it aims at minimizing the regret caused by both arms. This distinction poses a challenge for dueling bandit algorithms, necessitating a design that differs from the generalized linear bandit algorithm.
>
> ----
>
> **Q2**: Typos and minor issues:
> title of subsection 4.3 (sysmmetric -> symmetric)
> I may overlook. What is $\kappa$ in line 20 Algorithm 1 and Eqn. (4.3)?
>
> **A2**: Thanks for pointing out the typos. $\kappa$ is the same as $\kappa_\mu$. We have revised them.
>
> ----
>
> **Q3**: Is the variance of the noise equal to under the Bernoulli setting? In that case I feel the variance would be fully dependent on the arms selected in each round.
>
>
> **A3**: Yes, the noise equals $p_t(1-p_t)$ under the Bernoulli setting, and it depends on the arm being selected in each round. We study the problem in a general case where the arm set is time-varying and therefore the variance of arms can vary with respect to time and arms. Therefore, the variance $p_t(1-p_t)$ in each round is influenced by both the historical data and the potentially adversarial arm set $\mathcal{A}_t$ in round $t$. Our result indicates that the regret of the problem should depend on the variance of the selected arms, which aligns with our intuition that comparing high-variance dueling arms pose a greater challenge than comparing lower-variance dueling arms.
>
> ----
>
> [1] Zhao et al., 2023, Variance-dependent regret bounds for linear bandits and reinforcement learning: Adaptivity and computational efficiency, COLT

---

> > ### Comment · Reviewer_83pH · 2023-11-22
> > **Thanks a lot for your response**
> >
> > Thank you very much for your detailed responses. I acknowledge the problem setting in this work is new, and I really appreciate the presentation of this work. However, I still hold concerns on the novelty of this work, given the existing literature, especially SAVE.
> >
> > 1. Yes, the analysis of MLE in generalized linear model is more difficult. However, I feel it is already a well-studied problem. As the authors mention in their work, the bound (Eqn. 4.2) can be deduced by using the results from Zhao et al. 2023a. Therefore, there is no new contribution here.
> >
> > 2. Compared with SAVE, I feel the only difference of the algorithm lies in line 8 in the pseudocode, where the author uses $x_t + y_t$ instead of $x_t - y_t$ for the exploitation term since the regret bound depends on $x_t + y_t$. And reason can be easily observed in Appendix C method 3 where the deduction is also not technically difficult. Therefore, I feel the theory is a mild extension to SAVE and not very novel.
> >
> > Additional question: For the variance-aware contextual dueling bandit, the variance completely depends on the pulled arm $x_t$, while for the general variance-aware contextual bandit the variance can be fully independent. Do author think this extra information will bring in some more benefits for analysis? It would be better to have a discussion on this issue.

---

> > > ### Author Response · Authors · 2023-11-22
> > > **Response to Follow-Up Questions**
> > >
> > > Thank you for your additional questions and feedback. Let us address your questions as follows.
> > >
> > > 1.We acknowledge that the analysis of MLE in the generalized linear model is well studied, but the variance-aware analysis of MLE for generalized linear bandits is limited, marking the novelty of our work. In addition, we provide a rigorous proof for the MLE estimator in Appendix D, which fixes an overlooked issue in previous MLE analysis of generalized linear models. This part is also new and a novel result.
> > >
> > > Furthermore, In Eqn (4.2), the only thing we use from [1] is the Freedom-type concentration inequality for self-normalized martingales, while the analysis for our estimator is still new, and is not deduced from [1].
> > >
> > > ----
> > >
> > > 2.We acknowledge that our algorithm shares some spirit with SAVE. However, SAVE is an algorithm for linear contextual bandits, and it is not immediately clear how to extend it to generalized linear bandits, let alone dueling bandits. For dueling bandits, we need to select two competing arms $\mathbf{x} _ t$ and $\mathbf{y} _ t$. We would like to point out that in Line 8, the arm selection involves both $\mathbf{x} _ t + \mathbf{y} _ t$ and $\mathbf{x} _ t-\mathbf{y} _ t$, which cannot be deduced from SAVE. As a comparison, in Line 8 of SAVE, the selected arm is $\mathbf{a} = \text{argmax} \_ \mathbf{a} \langle \mathbf {a}$ $, \hat\theta \rangle + \beta ||\mathbf a|| _ {\Sigma}$. The selected arm cannot simultaneously represent both for $\mathbf{x}_t + \mathbf{y}_t$ and $\mathbf{x}_t - \mathbf{y}_t$, and therefore it’s not deduced from SAVE.
> > >
> > > The derivation in Appendix C method 3 is our contribution. While it may not seem that difficult to figure out, it represents a novel addition that has not been previously documented in the existing body of work.
> > >
> > > ----
> > >
> > > To your additional question, we consider a very general case where the variance can depend on the pulled arm, and we do not make any assumption on the prior knowledge of these variances. Therefore, we do not think we can obtain any extra information for our analysis, unless we make additional assumptions. We will add this discussion to the paper.
> > >
> > > ----
> > >
> > > [1] Zhao et al., 2023, Variance-dependent regret bounds for linear bandits and reinforcement learning: Adaptivity and computational efficiency, COLT

---

### Official Review · Reviewer_Kvxm · 2023-10-27

**Soundness:** 3 good
**Presentation:** 3 good
**Contribution:** 3 good
**Rating:** 6
**Confidence:** 4

**Summary:**

The paper addresses the problem of contextual dueling bandits, where the feedback is based on pairwise comparisons between arms. The authors focus on scenarios where the binary comparisons are generated from a GLM. The authors proposed VACDB algorithm that adapts to the variance of the pairwise comparisons, leading to potentially better performance in scenarios with varying levels of uncertainty. The variance-aware regret bound of order $O\left(\sqrt{d\sum_{t=1}^{T} \sigma_t^2} + d\right)$, which also aligns with intuitive expectations, reducing to $O(d)$ in deterministic scenarios. The authors validate their approach through experiments on synthetic data, demonstrating the advantages of VACDB over previous variance-agnostic algorithms.

**Strengths:**

- The primary contribution of the paper is the introduction of a new algorithm VACDB (Variance-Aware Contextual Dueling Bandits), which incorporates a SupLinUCB-type approach to handle the contextual information and provide a variance-aware regret bound.
- The regret bound $O\left(\sqrt{d\sum_{t=1}^{T} \sigma_t^2} + d\right)$ proposed by the authors provides a more nuanced performance measure that reflects the difficulty of the decision-making problem.
- Beyond the specific algorithm and regret bound, the paper also contributes to the theoretical understanding of generalized linear bandits, correcting an issue in the existing analysis of the MLE estimator.

Overall, I think the paper enhances our understanding of decision-making in dueling bandits scenarios by explicitly accounting for the uncertainty in pairwise comparisons, providing both theoretical insights and practical algorithms to address the challenge on variance-awareness algorithms.

**Weaknesses:**

- The paper primarily conducts experiments on synthetic data to validate the proposed algorithm. While this is a common practice, the performance of the algorithm in real-world scenarios might differ. To strengthen the paper, the authors could include experiments on real-world datasets, particularly those related to the applications mentioned like ranking, recommendation systems, or any human-interactive system, ensuring the practicality and robustness of the algorithm in diverse settings.
- The concept of a layered design for bandit algorithms, which the authors adopt in the VACDB algorithm, has been previously explored in the literature. Layered or epoch-based approaches are widely used in bandit algorithms to balance exploration and exploitation in stochastic settings. The paper could benefit from a more thorough discussion on how the layered design in VACDB specifically contributes to or differs from existing approaches.
- The VACDB algorithm relies on the computation of regularized MLE for parameter estimation. The computation of these estimators is known to grow in complexity with the number of iterations, potentially leading to scalability issues, especially in settings with a large number of arms, contexts, or iterations.

**Questions:**

- There is a minor typo on the second line of Introduction: "arm" -> ``arm"


================================================

**After Rebuttal:**

I appreciate the response from the authors, and I am going to keep my original rating.

---

> ### Author Response · Authors · 2023-11-21
> **Response to Reviewer Kvxm**
>
> Thank you for your positive feedback. We address your questions below.
>
> **Q1**: Performance of the algorithm on real-world datasets.
>
> **A1**: To showcase the performance of our algorithms in a real-world setting, we have added a new experiment in Appendix B. We use EventTime dataset [1]. In this dataset, $K = 100$ historical events are compared in a pairwise fashion by crowd-sourced workers. The data contains binary response indicating which one of the events the worker thinks precedes the other. There is no side information
> $
>     \mathcal A = \\{ \mathbf x_i, i \in [K] \\},
> $
> or the true parameter $ \theta^*$ readily available in the dataset.  Thus, we estimate them with pairwise comparison data. To achieve this, let $C_{ij}, i,j \in [K]$ be the number of times event $j$ precedes event $i$ labeled by the workers. The following MLE estimator is used:
> $$
> \mathop{\mathrm{argmax}} _  { \\{\mathbf{x} _ i \\}, \theta} \sum_{i \in [K]} \sum_{j \in [K]} C_{ij}
> \log
> (
>     \sigma((\mathbf x _ i - \mathbf x _ j)^\top\theta)
> ).
> $$
> With the estimated $\mathcal A$ and $\theta^*$, it is then possible to simulate the interactive process. We can see that after about $2500$ rounds, our algorithm starts to outperform $\texttt{MaxInP}$ in terms of cumulative regret.
>
> ----
>
> **Q2**: The paper could benefit from a more thorough discussion on how the layered design in $\texttt{VACDB}$ specifically contributes to or differs from existing approaches.
>
> **A2**: Thanks for your suggestion. We have provided a detailed discussion in Appendix A in our revision. Our approach is different in the elimination phase, where we construct our confidence set with a variance adaptive radius to determine the confidence radius (lines 10-12).  Therefore, besides the layer number, our algorithm also leverages the variance information in the exploration. Moreover, we construct a carefully chosen weight dependent on the layer  (line 15). It is used in the calculation of the regularized MLE estimator.
>
> ----
>
> **Q3**: Computation of the MLE estimators grows in complexity with the number of iterations.
>
> **A3**:  The MLE estimator is widely used in the literature of generalized linear bandit and dueling bandits, such as [2] and [3]. Although the calculation can be expensive in some cases, there are some literally tractable solutions to that. The regularized MLE can be formulated as a finite-sum offline optimization problem. For many widely used models, such as the Bradley-Terry-Luce (BTL) model, the regularized MLE is a strongly convex and smooth optimization problem. We can solve it using accelerated gradient descent [4] and SVRG [5], both of which achieve a linear rate of convergence. This can mitigate the scalability issues caused by the increasing number of iterations. The regularized MLE estimator can also be solved by an online learning algorithm such as in [6,7].
>
> ----
>
> **Q4**: There is a minor typo on the second line of Introduction: "arm" -> ``arm"
>
> **A4**: Thanks for pointing that out. We have revised it.
>
> ----
>
> [1] Zhang et al., 2016, Crowdsourced top-k algorithms: An experimental evaluation. VLDB.
>
> [2] Li et al., 2017, Provably optimal algorithms for generalized linear contextual bandits, ICML
>
> [3] Saha 2021, Optimal algorithms for stochastic contextual preference bandits, NeurIPS
>
> [4] Nestorov 2003, Introductory lectures on convex optimization: A basic course,
>
> [5] Johnson & Zhang, 2013, Accelerating Stochastic Gradient Descent using Predictive Variance Reduction, NeurIPS
>
> [6] Jun et al., 2017, Scalable generalized linear bandits: online computation and hashing. NeurIPS
>
> [7] Zhao et al., 2023, Optimal online generalized linear regression with stochastic noise and its application to heteroscedastic bandits, ICML

---

### Official Review · Reviewer_W8bc · 2023-10-31

**Soundness:** 3 good
**Presentation:** 3 good
**Contribution:** 3 good
**Rating:** 5
**Confidence:** 3

**Summary:**

This work studied the stochastic contextual dueling bandits. It proposed the VACDB algorithm, which is a new SupLinUCB-type algorithm. It provided a variance-aware bound on the regret of the proposed algorithm. This work presented a detailed review on existing literature and clarified the novelty of the proposed algorithm. It also evaluated the performance of the algorithm with numerical experiments.



==============

I appreciate the response from the author(s). I may increase the score if they further resolve my concern.

**Strengths:**

1. This paper is in general well organized and easy to follow.
1. The variance is considered in the dueling bandit setting.
1. I appreciate the detailed review on existing literature, and the clarification on the novelty of the proposed algorithm and the differences from existing algorithms (especially SupLinUCB-type ones).

**Weaknesses:**

1. The variance $\sum_{t=1}^T \sigma_t$ in the regret bound is a random variable. I think it would be much better to involve a term that indicates the variance of the instance in some sense but is not random in an expected bound.
    1. The appearance of $\sum_{t=1}^T \sigma_t$ indicates that even if we know $X_t$ for all arms and $\theta^*$, we may not know the value of the derived upper bound.
1. I wonder if it is possible to derive a lower bound for the problem. If not, may the author(s) clarify the analytical challenge?

**Questions:**

Please refer to the **Weaknesses** section.

---

> ### Author Response · Authors · 2023-11-21
> **Response to Reviewer W8bc**
>
> Thank you for your valuable feedback. We address your questions as follows.
>
> **Q1**: The variance in the regret bound is a random variable. I think it would be much better to involve a term that indicates the variance of the instance in some sense but is not random in an expected bound. We may not know the value of the derived upper bound when knowing the exact model.
>
> **A1**: Yes, the randomness of the variance is from the arm being chosen in each round, which further depends on historical data. The setting we study is quite general where the arm set is time-varying and therefore the variance of arms can vary with respect to time and arms. That’s why the regret depends on a random variable representing the variance of the chosen arms in each round. We view this as a strength rather than a weakness of our regret bound.
>
> When we restrict our setting to a special case where the variances for all the pairwise comparisons are identical, the randomness of $\sigma_t^2$ can be removed in the regret bound. But this is similar to the worst-case regret implied by our variance-aware regret.
>
> ----
>
> **Q2**: Is it possible to derive a lower bound for the problem?
>
> **A2**: We study this problem in a general setting where the variance of instances can vary over time and across different arms. In this context, a variance-dependent lower bound is difficult and not found even in the literature on stochastic bandit settings. Deriving that lower bound stands as an open problem. The proof is challenging especially when the variance can change over time. For example, consider a scenario where initially there is no variance, but later, substantial variance emerges. In such cases, accurate early exploration could make correct policies, effectively mitigating the impact of later variance. However, the variance-dependent regret bound in the literature does not inherently possess this characteristic. In general, we are not sure if it is possible, and leave it as an interesting future work.
>
> Although getting a lower bound in the general case is hard if not impossible, we can show our optimality in two special cases:
>
>  (1) when we consider the worst case, the variance is upper bounded by $1/4$, our result matches the $d\sqrt{T}$ lower bound proved in [1]; and (2) when the preference is deterministic, our result is $\tilde O(d)$, which is also nearly optimal.
>
> ----
> [1] Bengs et al., 2022, Stochastic contextual dueling bandits under linear stochastic transitivity models, ICML

---

> > ### Comment · Reviewer_W8bc · 2023-11-22
> > **Thanks for the response**
> >
> > 1. As the author(s) said that 'When we restrict our setting to a special case where the variances for all the pairwise comparisons are identical, the randomness of $\sigma_t^2$ can be removed in the regret bound.', I am curious what is this bound.
> > 2. If we consider the worst case 'the variance is upper bounded by $1/4$', do the upper and lower bounds match regarding terms except for $d$?
> >
> > I appreciate that the author(s) include these two points in the main paper.

---

> > > ### Author Response · Authors · 2023-11-22
> > > **Response to Follow-Up Questions**
> > >
> > > Thank you for your additional questions and suggestions.
> > >
> > > For the first question, when we restrict our setting to the special case with uniform variances for all pairwise comparisons, i.e., $\sigma_t^2 = \sigma^2$ for all $t$, our upper bound becomes $\tilde O(\sigma d \sqrt T)$. This results in a regret bound that does not depend on the random variable $\sigma_t^2$.
> > >
> > > For the second question, in the worst case, when the variance is upper bounded by ¼, our upper bound becomes $\tilde O(d\sqrt{T})$, which matches the lower bound $\Omega (d\sqrt{T})$ [1] up to logarithmic factors.
> > >
> > > We have added these two points in Remarks 5.3 and 5.4 in our revised paper.
> > >
> > > [1] Bengs et al., 2022, Stochastic contextual dueling bandits under linear stochastic transitivity models, ICML

---

### Official Review · Reviewer_2YSg · 2023-11-01

**Soundness:** 3 good
**Presentation:** 3 good
**Contribution:** 3 good
**Rating:** 8
**Confidence:** 2

**Summary:**

The paper studies contextual dueling bandits, where binary comparison is generated from a generalized linear model. The work proposes a new framework to obtain variance aware regret guarantees. This work further provides empirical results comparing the algorithm against baseline.

**Strengths:**

The problem setup is well laid out and easy to follow with the precisely required assumptions.

**Weaknesses:**

The paper is not self contained and requires reader to go through multiple papers for example in section 4.2.

**Questions:**

No questions.

---

> ### Author Response · Authors · 2023-11-21
> **Response to Reviewer 2YSg**
>
> Thank you for your strong support. We address your question as follows.
>
> **Q1**: The paper is not self contained and requires the reader to go through multiple papers for example in section 4.2.
>
> **A1**: Thanks for your suggestion. In the main text, we have added Section 4.1 to give an overview of the algorithm and introduce the multi-layered structure in more detail. We have also added Appendix A to provide a detailed introduction to our algorithm and a comparison with previous works.

---

### Meta-Review · Area_Chair_dbfM · 2023-12-09

**Metareview:**

A regret bound that scales with cumulative variance of the selected arms for generalized linear bandits is novel and interesting. Reviewers generally agree on this point and I share the view that the authors' response towards the reviewers' questions provide a reasonable argument confirming this work's merit (such as scaling with a changing variance is not a restriction). The bandits community would be interested in learning the new effects produced by variance.

**Justification For Why Not Higher Score:**

Tightness of the results remain open, leaving more to be desired.

**Justification For Why Not Lower Score:**

N/A

---

### Decision · Program_Chairs · 2024-01-16

Accept (poster)